# HeraSys: Collaborative Serving of Multiple LLM Workflows via Fine-Grained End-to-End Optimization

**Size Li** [1 2]  **Zhiqing Tang**[✉ 2 1]  **Hongrui Liang** [1 2]  **Jianxiong Guo** [2 3]  **Jiong Lou** [4]  **Tian Wang** [2]  **Weijia Jia** [2 3]

## Abstract

The proliferation of Large Language Models (LLMs) has shifted serving systems from processing isolated requests to orchestrating high-concurrency, multi-tenant agentic workflows. However, existing solutions typically prioritize intra-workflow optimization, largely neglecting the significant potential for inter-workflow optimization. In this paper, we propose HeraSys, an LLM serving system designed to optimize the end-to-end performance of concurrent workflows. Through fine-grained orchestration, HeraSys eliminates cross-workflow computational redundancy via structural node merging and reuse. Furthermore, HeraSys introduces a load-aware joint scheduling policy that dynamically manages execution order by evaluating both inter- and intra-query priorities. By integrating a resource skewing mechanism with adaptive batching and pipeline decomposition, HeraSys effectively mitigates tail latency while maintaining low average latency, thereby substantially improving system throughput. Extensive experiments demonstrate that HeraSys reduces P99 latency by up to $2.17\times$ and increases serving throughput by up to $1.85\times$ under strict latency guarantees.

## 1. Introduction

The rapid development of Large Language Models (LLMs) has moved modern application development from simple single-turn conversational interfaces to complex agentic workflows composed of multiple LLM calls and external

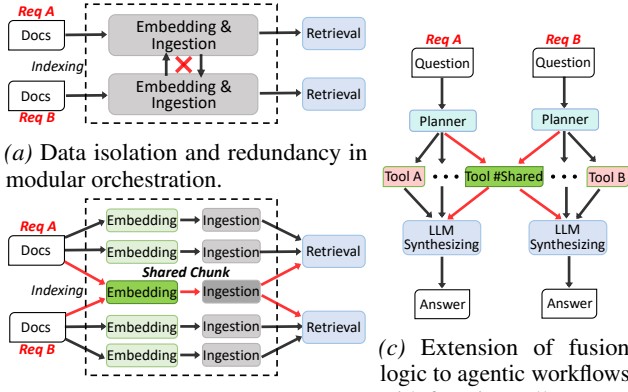

*(a)* Data isolation and redundancy in modular orchestration.

*(b)* Fusion opportunities exposed by fine-grained orchestration.

*(c)* Extension of fusion logic to agentic workflows with function calls.

*Figure 1.* Computational redundancy and fusion opportunities.

tools. To execute these tasks, applications are typically represented as Directed Acyclic Graphs (DAGs) containing interdependent steps. Currently, serving systems primarily optimize performance in a single dimension: either focusing on the dependencies between LLM inference tasks (Kwon et al., 2023; Lin et al., 2024; Zhong et al., 2024; Shen et al., 2026) or the orchestration logic within a single workflow (Chase & LangChain Team, 2024; 2025; Wu et al., 2024b). These approaches effectively optimize individual requests but do not address the broader context of system-wide execution.

In multi-tenant settings, however, serving systems typically treat concurrent workflows as independent instances. This approach overlooks critical application-layer information shared across requests. While some recent works attempt to mitigate this issue, they generally lack a holistic view of the global context, failing to effectively address the inherent redundancies and resource contention in concurrent execution (Mei et al., 2025; Liu et al., 2025b; Lin et al., 2024; Luo et al., 2025; Shen et al., 2025). We observe that by breaking this request-level isolation to jointly optimize cross-workflow computational redundancy and global coordinated scheduling, significant improvements can be realized in end-to-end performance.

The first challenge is how to mitigate the substantial over-

[1]Faculty of Arts and Sciences, Beijing Normal University, Zhuhai, China [2]Institute of Artificial Intelligence and Future Networks, Beijing Normal University, Zhuhai, China [3]Guangdong Key Lab of AI & Multi-Modal Data Processing, Beijing Normal-Hong Kong Baptist University, Zhuhai, China [4]School of Computer Science, Shanghai Jiao Tong University, Shanghai, China. Correspondence to: Zhiqing Tang <zhiqingtang@bnu.edu.cn>.

*Proceedings of the $43^{rd}$ International Conference on Machine Learning*, Seoul, South Korea. PMLR 306, 2026. Copyright 2026 by the author(s).

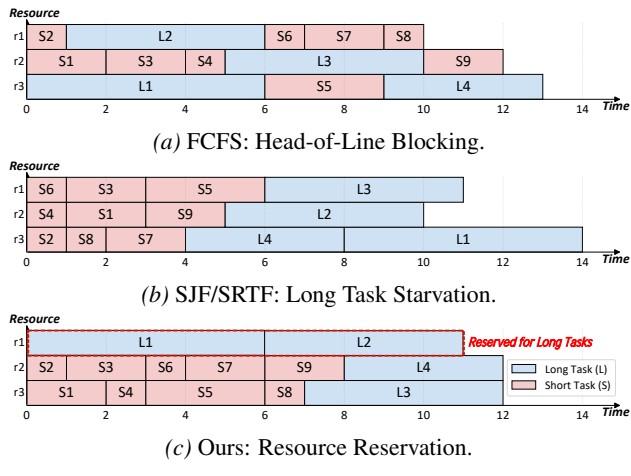

*(a)* FCFS: Head-of-Line Blocking.

*(b)* SJF/SRTF: Long Task Starvation.

*(c)* Ours: Resource Reservation.

*Figure 2.* Mitigating blocking and starvation via resource skewing.

head caused by treating workflows in isolation, which obscures critical task redundancy (Shen et al., 2025; Gim et al., 2024). For example, in production LLM applications, multiple concurrent queries often involve embedding and retrieving the same batch of common document chunks, as shown in Figure 1b. Such fine-grained data redundancy allows for fusion and deduplication. Similarly, in agentic workflows, Figure 1c illustrates that different agents may need to invoke the same tools or process similar prefix contexts (Abhyankar et al., 2024). However, as shown in Figure 1a, existing modular designs lack a global view and fail to detect these overlaps, forcing the system to execute redundant tasks. This not only increases system load but also limits system throughput. Consequently, exploiting these fine-grained structural commonalities via cross-workflow fusion presents a promising avenue to eliminate computational waste.

The second challenge is how to balance latency and fairness in scheduling, an objective complicated by workload heterogeneity. Real-world workloads typically mix short tasks (e.g., simple QA) and long tasks (e.g., long document analysis, multi-round iterative reasoning). In this setting, simple First-Come-First-Served (FCFS) or Round-Robin scheduling often leads to Head-of-Line (HOL) Blocking: as shown in Figure 2a, a long-running task may occupy resources for an extended period, delaying numerous subsequent short tasks. Conversely, as seen in Figure 2b, strictly prioritizing short tasks (SJF/SRTF) may cause resource starvation for long tasks, resulting in unfairness (Luo et al., 2025; Wu et al., 2024a). To address this, we find that combining resource reservation for long tasks with prioritization for short tasks prevents starvation and balances performance with Quality of Service (QoS), as demonstrated in Figure 2c.

In this paper, we propose HeraSys, a fine-grained end-to-end serving system for multi-workflow scenarios (§2.1). HeraSys adopts a two-layer optimization architecture: (I)

The Pre-Execution Graph Fuser identifies and fuses redundant subgraphs across queries before execution, generating a more efficient graph. (II) The Runtime Load-Aware Joint Scheduler dynamically prioritizes incoming query tasks, biases resource allocation towards potential bottlenecks, and adjusts batch sizes and pipeline decomposition granularity based on system load. This ensures the system maintains high efficiency and meets service requirements under dynamic load changes. We implement HeraSys on a server node equipped with NVIDIA RTX 4090 GPUs, using vLLM as the inference backend. We evaluate HeraSys against representative baselines, including LangGraph (Chase & LangChain Team, 2025), LlamaIndex (Liu & LlamaIndex Team, 2025), and the state-of-the-art system Ayo (Tan et al., 2025), using diverse workloads such as RAG, Web Search, and mixed scenarios. Extensive experiments validate the effectiveness of our design, showing that HeraSys delivers substantial performance gains in both end-to-end latency and system throughput compared to state-of-the-art baselines.

The main contributions of this paper are as follows:

1. We design a dual-layer optimization architecture to address the bottlenecks of existing serving systems in multi-workflow scenarios. This architecture overcomes the traditional request-level isolation, achieving joint optimization of resource utilization and scheduling efficiency through the integration of pre-execution graph fusion and runtime dynamic scheduling.

2. We propose a cross-query graph node reuse mechanism that identifies and eliminates redundant computational subgraphs across workflows, significantly reducing computational overhead (§3.3).

3. We propose a load-aware joint scheduling strategy. By integrating dynamic long/short query classification, biased resource allocation, and adaptive batching, this strategy effectively reduces the end-to-end latency of query processing (§3.4).

4. We implement HeraSys and demonstrate the effectiveness of our approach through extensive experimental comparisons with state-of-the-art systems.

**Conflict of Interest Disclosure.** The authors declare no financial conflicts of interest related to this work.

## 2. Related Work

### 2.1. Fine-Grained Workflow Orchestration

Early LLM frameworks used modular abstractions (Figure 3a) to simplify orchestration, obscuring upper-layer control flow and data dependencies from serving systems.

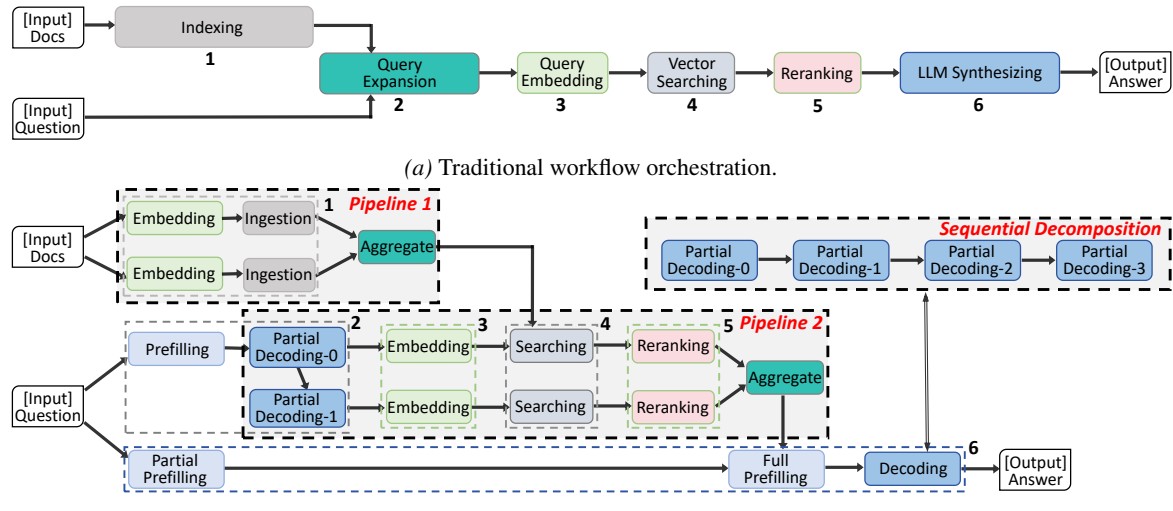

*(a)* Traditional workflow orchestration.

*(b)* Fine-grained workflow orchestration.

*Figure 3.* Structural comparison of orchestration paradigms. Fine-grained orchestration decomposes modular architectures into fine-grained primitive nodes, such as splitting Indexing into Embedding and Ingestion, and splitting LLM Prefilling into partial and full phases, exposing greater parallelism and optimization space. **Additionally, HeraSys decomposes LLM Decoding into sequential sub-nodes and dynamically regulates the pipeline decomposition granularity.**

This limited end-to-end optimization (Liu & LlamaIndex Team, 2025; Chase & LangChain Team, 2024; 2025; Wu et al., 2024b). To address the bottlenecks of modular encapsulation, recent research has moved towards fine-grained orchestration. As shown in Figure 3b, Ayo (Tan et al., 2025) proposes a task-primitive-based approach, decoupling modular tasks into general computational nodes to enable better parallelism. Similarly, other works focus on optimizing data pipelines (Lin et al., 2024; Shen et al., 2025; Hong et al., 2024; Qian et al., 2024; Khattab et al., 2024). Although these systems improve intra-workflow efficiency by adjusting orchestration granularity, they often miss global optimization opportunities across workflows.

### 2.2. Multi-level Redundancy Elimination

Eliminating redundancy is key for system throughput. At the inference level, vLLM (Kwon et al., 2023) and SGLang (Zheng et al., 2024) use block-based memory and radix tree-based prefix caching to enable KV cache reuse. Approaches like GPTCache (Bang, 2023) and Prompt Cache (Gim et al., 2024) explore reuse based on semantic similarity or schema modularity. Other research targets cross-instance distributed deduplication, enabling low-latency sharing via distributed cache pools or encoding techniques (Qin et al., 2025; Yao et al., 2025; Shen et al., 2025; Liu et al., 2024; 2025a). While these works reduce redundancy at various levels, they fail to detect the complex global graph topology within workflows and cannot identify structural redundancies before execution.

### 2.3. Scheduling for Heterogeneous Workflows

LLM workflows are highly heterogeneous. While early inference schedulers introduced techniques such as iteration-level scheduling to reduce batching-induced blocking (Yu et al., 2022; Wu et al., 2024a; Leviathan et al., 2023), they struggle to handle complex workflows. To address this, DistServe (Zhong et al., 2024), Sarathi-Serve (Agrawal et al., 2024), and Splitwise (Patel et al., 2024) decouple inference phases to improve resource utilization. Autellix (Luo et al., 2025) analyzes HOL blocking and proposes scheduling based on attained service time. Hermes (Liu et al., 2025b) handles volatility using Probabilistic Demand Graphs, and Helix (Mei et al., 2025) optimizes cluster scheduling via max-flow formulation. However, under dynamic workloads, these methods are often limited by modeling overhead or bias and lack fairness for long-tail tasks. Thus, a lightweight, feedback-driven mechanism is needed to reduce tail latency and ensure fairness.

## 3. Method

### 3.1. Problem Formulation

We formulate the multi-tenant query scheduling and resource allocation problem as an online optimization task. At any time $t$, the system receives and processes a set of arriving queries $\mathcal{Q}_t$.

**Workload Model.** Each query $q \in \mathcal{Q}_t$ is modeled as a DAG $G_q = (\mathcal{V}_q, \mathcal{E}_q)$, where $\mathcal{V}_q$ is the set of task nodes and $\mathcal{E}_q$ represents their sequential dependencies. For each node $v \in \mathcal{V}_q$, we define the following attributes: $w_v$ repre-

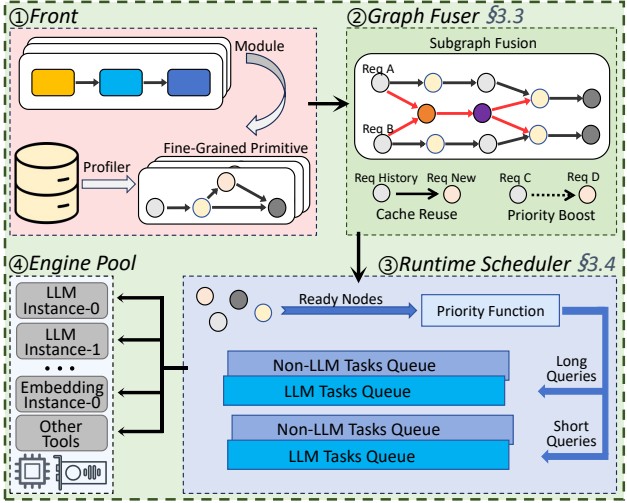

*Figure 4.* System overview of HeraSys.

sents the workload estimation (specifically, input text size or FLOPs for non-LLM nodes, and expected token count for LLM nodes); $p_v$ specifies the set of executable device types, e.g., Embedding or LLM instances; $d_v$ denotes the reverse topological depth, defined as the distance from the node to the sink; and $s_v$ indicates the splittability upper bound, allowing the node to be parallelly split into $k$ sub-nodes.

**Resource Model and Decision Variables.** Let $\mathcal{R}$ denote the set of available system resources. Each resource $r \in \mathcal{R}$ is characterized by its type $\tau(r)$, parallelism capacity $cap_r$, and batching limit $B_r$. At each decision step, the system determines four variables: $x_{v,r} \in \{0, 1\}$ indicates whether to schedule node $v$ on resource $r$; $b_{v,r} \in \mathbb{N}^+$ represents the batch size; $a_{v,r}$ denotes the resource share allocation; and $k_v \in [1, s_v]$ is the pipeline decomposition quantity. As illustrated in Figure 3b, $k_v$ allows the system to decompose linear task chains (e.g., Embedding $\rightarrow$ Search $\rightarrow$ Rerank) into $k$ parallel sub-pipelines along the data dimension to exploit fine-grained parallelism.

**Optimization Objective.** Our objective is to balance average latency with fairness for long queries. We define the global objective function as:

$$\min \quad \mathcal{J} = \lambda \cdot \frac{1}{|\mathcal{Q}_t|} \sum_{q \in \mathcal{Q}_t} C_q + (1 - \lambda) \cdot \max_{q \in \mathcal{Q}_t} C_q, \quad (1)$$

where $C_q$ denotes the end-to-end completion time of query $q$, and $\lambda \in [0, 1]$ is a coefficient that balances the average and maximum latency.

### 3.2. System Overview

As illustrated in Figure 4, HeraSys comprises four primary components.

**Declarative Frontend and Profiler.** This component parses user-defined DAG workflows into fine-grained task nodes ($\mathcal{V}_q$). It uses a profiling library to estimate the workload $w_v$ and remaining time $S_q$ to guide scheduling.

**Pre-Execution Graph Fuser (§3.3).** This module identifies redundant subgraphs across queries and eliminates duplicate computation through node fusion. It also raises the priority of shared nodes to accelerate query execution.

**Runtime Scheduler (§3.4).** The scheduler dynamically prioritizes tasks using a joint priority function $\pi_v$ and resource skewing to prevent starvation. It adaptively adjusts the batch size $b_{v,r}$ and pipeline decomposition $k_v$ based on system load.

**Backend Engine Pool.** This layer manages heterogeneous resources (e.g., LLMs, Embedding models) to execute dispatched tasks. It returns results to the scheduler to trigger subsequent dependencies.

### 3.3. Pre-Execution Optimization: Semantic Graph Fusion

**Node Semantic Signature.** To identify redundant computations, we use the semantic signature as a unique identifier to determine node equivalence. For any node $v$, its signature is a hash summary derived from its operator semantics and input context:

$$Sig(v) = \mathcal{H}\left(\tau_v, \Theta_p, \mathcal{D}_{in}, \bigoplus_{u \in \text{Pre}(v)} Sig(u)\right), \quad (2)$$

where $\tau_v$ represents the operator type; $\Theta_p$ contains parameters affecting the result (e.g., Top-K for retrieval); $\mathcal{D}_{in}$ summarizes the input data that determine the output of the node; and $\bigoplus$ represents the ordered combination of predecessor signatures. For example, in retrieval or embedding nodes, $\mathcal{D}_{in}$ is derived from the normalized query text or the document chunk content involved in that node. This addressing mechanism allows the system to map logically independent but computationally equivalent tasks to identical signatures across different queries.

**Dynamic Fusion and Reuse Mechanism.** The system implements two complementary deduplication strategies based on node arrival time.

(I) Topological Fusion within Time Window. For concurrent queries submitted within a short window, the system identifies redundancy via signature matching. When a node $v_n$ in a new query matches a node $v_e$ in the ready queue, the fuser removes $v_n$ from the execution graph and cancels its resource allocation. It then performs DAG edge redirection, modifying the topology to connect the downstream dependencies of $v_n$ to the existing node $v_e$. This physically fuses multiple logical queries to share a single execution instance. Upon completion, the output of $v_e$ flows to the downstream nodes of all related queries.

(II) Asynchronous Cache Reuse. For requests outside the time window, we employ asynchronous reuse. If a new node $v_n$ matches a completed node $v_h$, the system directly retrieves the result, skipping scheduling and execution. If $v_n$ matches a node $v_r$ that is currently executing, $v_n$ is not submitted to the engine but waits for $v_r$ to finish. Once $v_r$ completes, the system shares the result with $v_n$. This mechanism operates on the fine-grained workflow graph. For LLM operators, shared prefixes can be represented as decomposed prefill nodes, so matching such nodes enables HeraSys to redirect downstream dependencies or reuse completed results. Beyond LLM prefix reuse, the same graph-level mechanism also covers retrieval, reranking, tool execution, and other intermediate subgraphs.

**Reuse-Driven Priority Adjustment.** Graph fusion not only alters the graph structure but also influences scheduling. A highly reused node is not treated as a standard single task, but as a critical node enabling multiple downstream paths. To reflect this, we introduce a reuse gain to adjust the scheduling priority. For a fused node $v^*$, its priority contribution $\phi_{v^*}$ is defined as the sum of the contributions of all merged logical nodes (the definition of $\phi_v$ is detailed in §3.4):

$$\phi_{v^*} = \sum_{v \in \mathcal{M}(v^*)} \phi_v. \tag{3}$$

This ensures that nodes shared by multiple queries receive higher priority. By prioritizing these common sub-tasks, the system accelerates the progress of multiple concurrent workflows simultaneously.

### 3.4. Runtime Optimization: Load-Aware Joint Scheduling

**Query Feature Extraction.** The system uses two key runtime metrics to guide scheduling decisions.

(I) Dynamic Remaining Time Estimation ($\widehat{S}_q$). We combine offline profiling with execution progress to predict remaining time. To enable fine-grained preemption and accurate tracking, we decompose long decoding operators into sequential sub-nodes (Figure 3b). For a query $q$, $\widehat{S}_q$ is the sum of expected execution times for all nodes in the current unfinished subgraph. To account for LLM inference variability, we introduce a runtime correction factor $\mathcal{F}$. If actual execution exceeds expectations, $\mathcal{F}$ increases the estimate (Luo et al., 2025). The estimate at time $t$ is:

$$\widehat{S}_q(t) = \mathcal{F}\left(\frac{A_q(t)}{\sum_{v \in \mathcal{V}_{done}^q(t)} \bar{T}_v + \epsilon_T}\right) \cdot \sum_{v \in \mathcal{V}_{remain}^q(t)} \bar{T}_v \tag{4}$$

where $A_q(t)$ is the attained service time (accumulated execution time) for query $q$, and $\bar{T}_v$ denotes the predicted runtime of node $v$. Since $\widehat{S}_q$ is aggregated from node-level predictions, stages such as prefill, decoding, embedding, and retrieval contribute according to their predicted runtimes,

while $\mathcal{F}$ updates the estimate based on online execution progress. This metric allows the scheduler to identify short tasks nearing completion, prioritizing them to reduce average latency.

(II) Node Contribution ($\phi_v$). We introduce $\phi_v$ to quantify the contribution of node $v$ to query progress. This metric considers three topological features: downstream remaining workload, path length, and the number of parallel branches enabled. The node contribution is a weighted sum of these features:

$$\phi_v = \sum_{u \in \mathcal{N}_v} (\omega_1 \cdot w_u + \omega_2 \cdot d_u) + \omega_3 \cdot |\mathcal{N}_v|, \tag{5}$$

where $N_v$ is the set of successor nodes. Nodes with high $\phi_v$ are prioritized to accelerate the progress of the current task stream.

**Dynamic Partitioning and Resource Skewing.** To prevent long-task starvation, the system adopts a partitioning method based on load distribution.

(I) Long/Short Query Partitioning. The scheduler monitors the distribution of $\widehat{S}_q$ for active queries, calculating the mean $\mu_s$ and standard deviation $\sigma_s$. We define a dynamic threshold $\tau_{load} = \mu_s + \eta \cdot \sigma_s$. If $\widehat{S}_q > \tau_{load}$, the query is marked as a long query. Its computationally intensive nodes are marked as heavy nodes ($y_v = 1$), while others remain normal ($y_v = 0$).

(II) Resource Skewing. To address starvation, the scheduler implements resource skewing. The scheduler maintains separate candidate queues for short and long queries on each resource type. It enforces a minimum resource reservation, ensuring that at least a specific proportion of resources is allocated to heavy nodes of long queries:

$$\sum_{v:y_v=1} a_{v,r} \geq \rho \cdot cap_r. \tag{6}$$

This reservation isolates resource competition between tasks with different characteristics. It maintains low latency for short queries while guaranteeing progress for long tasks, preventing unfairness and tail latency degradation.

**Joint Priority Scheduling.** Combining these metrics, the scheduler uses a joint priority algorithm to order nodes in the ready queue. The algorithm balances latency, structural efficiency, and fairness via a composite priority function $\pi_v$:

$$\pi_v = \frac{\alpha}{\widehat{S}_{q(v)} + \epsilon} + \gamma \cdot \phi_v + \delta \cdot y_v, \tag{7}$$

where the first term implements dynamic Shortest Remaining Time First (SRTF) to reduce average latency (Schrage, 1968). The second term incorporates topological information, prioritizing critical nodes that enable downstream work to improve efficiency. The third term prevents starvation by

boosting the priority of heavy nodes in long tasks, working alongside resource skewing.

**Adaptive Execution Adjustment.** The system employs a load-adaptive mechanism to optimize performance under varying pressure. We address the global objective function:

$$\min \quad \mathcal{J}_t = \lambda_t \cdot \frac{1}{|\mathcal{Q}_t|} \sum_{q \in \mathcal{Q}_t} C_q + (1 - \lambda_t) \cdot \max_{q \in \mathcal{Q}_t} C_q, \quad (8)$$

where $\lambda_t \in [0, 1]$ is dynamically determined by system load. The scheduler switches between two states:

(I) In high-load intervals ($\lambda_t \to 1$), the scheduler increases the batching window $b_{v,r}$ to aggregate larger batches for higher throughput. Simultaneously, it reduces pipeline splits $k_v$ to minimize scheduling overhead.

(II) In low-load intervals ($\lambda_t \to 0$), the scheduler reduces the batching window to execute tasks immediately. It also increases $k_v$, decomposing nodes into micro-batches to exploit data parallelism and reduce end-to-end latency.

## 4. Experiments

### 4.1. Experimental Setup

**Testbed.** We implement the HeraSys prototype on Ray (Moritz et al., 2018). For the Graph Fuser, we identify structural relationships using reverse traversal and compute node semantic hashes via SHA-256 (NIST & Dang, 2015). For the Runtime Scheduler, we use Ray metrics to monitor system load, manage pending tasks using priority queues, and implement global scheduling via Ray's Actor model. We run experiments on a server node with 4 NVIDIA GeForce RTX 4090 GPUs. To ensure consistency across backends, we use vLLM (Kwon et al., 2023) for all systems. We use Llama-3-8B (Meta LLaMA Team, 2024) as the core LLM, bge-large-en-v1.5 (Xiao et al., 2024) for embedding (stored in PostgreSQL (Momjian & PostgreSQL Global Development Group, 2025) with pgvector (Kane, 2025)), and bge-reranker-large for reranking. For external retrieval, we integrate the Google Search API.

**Baselines.** To evaluate HeraSys, we compare it against three representative systems:

- **LlamaIndex (Liu & LlamaIndex Team, 2025):** A data framework connecting LLMs with external data, LlamaIndex optimizes indexing and retrieval pipelines. It provides a complete toolchain from data ingestion to query routing. At runtime, it follows predefined logic, sequentially executing steps such as retrieval, reranking, and response synthesis.

- **LangGraph (Chase & LangChain Team, 2025):** A modular agent orchestration framework, LangGraph models workflows as execution graphs of LLM calls and tool executions, managing context flow via a state machine. At runtime, LangGraph creates an independent state graph for each request and schedules execution in topological order within its isolated scope.

- **Ayo (Tan et al., 2025):** Ayo is a state-of-the-art fine-grained orchestration system. Ayo proposes a decoupled architecture based on "Task Primitives", decomposing LLM applications into standardized operators. It implements data parallelism within statically defined pipelines, supporting heterogeneous resources.

**Workloads.** We construct four workloads with distinct characteristics based on public datasets to simulate multi-tenant traffic.

- **Standard RAG:** The system embeds user queries and documents using bge-large-en-v1.5, followed by vector retrieval in PostgreSQL. We feed the top-12 relevant chunks into the core LLM (Llama-3-8B) to synthesize the answer. The query set is sampled from the MS MARCO dataset (Bajaj et al., 2018).

- **RAG with Rerank:** This extends the standard RAG workflow to enhance accuracy. A reranker (bge-reranker-large) scores the initial top-12 chunks. The system selects the top-8 chunks based on these scores for generation.

- **Web Search Agent:** This simulates a multi-step agentic task. The LLM rewrites the user intent into a search query, invokes the Google Search API, and synthesizes results to generate a response. Test queries are sampled from TruthfulQA (Lin et al., 2022).

- **Mixed Workload:** This consists of 75% RAG tasks (mixed standard and rerank) and 25% Web Search tasks to simulate a heterogeneous scenario where tasks with different patterns coexist.

To mimic production traffic, we submit requests using a multi-threaded client to simulate concurrent access. We vary concurrency across 6, 30, 100, and 300 to cover light to saturated loads. We inject 5%–15% duplicate queries targeting hot documents following a Zipfian distribution (Breslau et al., 1999) to evaluate fusion. For resource allocation, we deploy two vLLM instances on two GPUs, and run 3 Embedding and 3 Reranking instances on the remaining GPUs to reduce bottlenecks and maximize performance. In the main experiments, HeraSys uses a fixed scheduler configuration, with $\alpha = 1.0$, $\gamma = 0.3$, and $\delta = 0.5$ for the joint priority function in Equation 7, and $\rho = 0.2$ for resource skewing in Equation 6. Finally, we optimize configurations for all baselines, including batch size tuning, to ensure fair comparison under latency constraints.

*Table 1.* End-to-end tail latency performance ($P_{95}$ and $P_{99}$) across different workflows and concurrency levels. The Speedup row lists the improvement of HeraSys in each scenario.

| Workflow | System | Concurrency 6 | | Concurrency 30 | | Concurrency 100 | | Concurrency 300 | |
|---|---|---|---|---|---|---|---|---|---|
| | | $P_{95}$ (s) | $P_{99}$ (s) | $P_{95}$ (s) | $P_{99}$ (s) | $P_{95}$ (s) | $P_{99}$ (s) | $P_{95}$ (s) | $P_{99}$ (s) |
| RAG w/ Rerank | LangGraph | 4.57 | 4.58 | 12.02 | 12.17 | 38.09 | 39.80 | 98.85 | 115.50 |
| | LlamaIndex | 4.35 | 4.36 | 11.45 | 11.59 | 36.28 | 37.90 | 94.14 | 97.31 |
| | Ayo | 4.00 | 4.01 | 10.42 | 10.55 | 32.65 | 34.11 | 83.78 | 86.61 |
| | **HeraSys** | **3.80** | **3.83** | **9.04** | **9.39** | **26.53** | **27.21** | **58.39** | **63.07** |
| | **Speedup** | **1.20×** | **1.20×** | **1.33×** | **1.30×** | **1.44×** | **1.46×** | **1.69×** | **1.83×** |
| RAG w/o Rerank | LangGraph | 3.88 | 3.89 | 10.22 | 10.34 | 32.38 | 33.83 | 84.02 | 103.39 |
| | LlamaIndex | 3.70 | 3.71 | 9.73 | 9.85 | 30.84 | 32.21 | 80.02 | 82.71 |
| | Ayo | 3.40 | 3.41 | 8.86 | 8.96 | 27.75 | 28.99 | 71.22 | 73.62 |
| | **HeraSys** | **3.36** | **3.37** | **8.08** | **8.37** | **23.84** | **24.67** | **54.59** | **59.21** |
| | **Speedup** | **1.15×** | **1.15×** | **1.26×** | **1.24×** | **1.36×** | **1.37×** | **1.54×** | **1.75×** |
| Web Search | LangGraph | 6.22 | 6.23 | 16.37 | 16.57 | 51.88 | 54.20 | 134.62 | 170.78 |
| | LlamaIndex | 5.65 | 5.67 | 14.88 | 15.07 | 47.16 | 49.27 | 122.38 | 126.50 |
| | Ayo | 5.20 | 5.21 | 13.93 | 14.10 | 43.77 | 45.72 | 112.59 | 116.38 |
| | **HeraSys** | **5.35** | **5.37** | **12.05** | **13.10** | **32.03** | **37.74** | **69.58** | **88.12** |
| | **Speedup** | **1.16×** | **1.16×** | **1.36×** | **1.26×** | **1.62×** | **1.44×** | **1.93×** | **1.94×** |
| Mixed Workload | LangGraph | 5.22 | 5.23 | 13.73 | 13.90 | 43.50 | 45.44 | 112.87 | 137.89 |
| | LlamaIndex | 4.74 | 4.75 | 12.48 | 12.63 | 39.55 | 41.31 | 102.61 | 106.07 |
| | Ayo | 4.36 | 4.37 | 11.36 | 11.50 | 35.99 | 37.18 | 91.33 | 94.40 |
| | **HeraSys** | **4.06** | **4.08** | **9.60** | **10.11** | **25.61** | **27.68** | **58.16** | **63.52** |
| | **Speedup** | **1.29×** | **1.28×** | **1.43×** | **1.37×** | **1.70×** | **1.64×** | **1.94×** | **2.17×** |

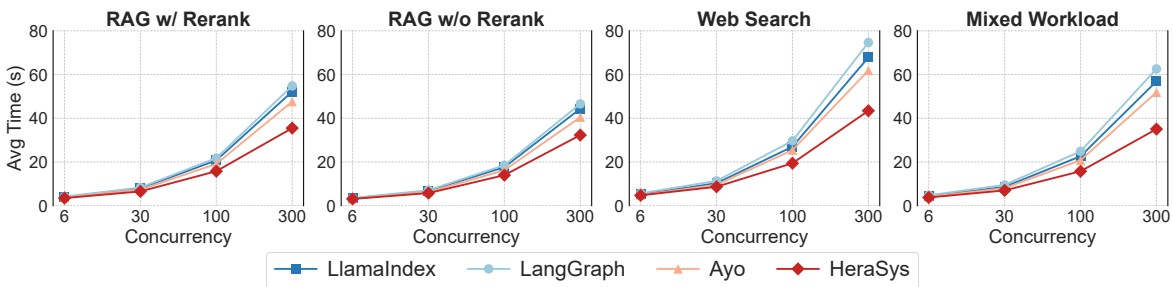

*Figure 5.* End-to-end average latency comparison across different workloads under varying concurrency levels.

## 4.2. End-to-End Latency Performance and Fairness

We first evaluate the end-to-end latency performance of HeraSys compared to baseline systems under different workloads and concurrency levels. As shown in Table 1 and Figure 5, HeraSys outperforms all baselines across all metrics.

**Average Latency.** Figure 5 shows that HeraSys reduces latency by 35.2% and 30.7% in the two RAG workflows (w/ and w/o Rerank), respectively, compared to the baselines. This is due to the SRTF strategy in HeraSys, which reduces average latency. In contrast, other baselines ignore real-time task duration, even though Ayo optimizes data pipelines through fine-grained task primitives. Notably, HeraSys achieves larger gains in the Web Search workflow, reducing average latency by 41.8%. Since this scenario calls external search APIs, other baselines suffer from blocking waits due to network I/O and API limits. HeraSys uses the Graph Fusion mechanism to fuse and reuse redundant queries, thereby reducing external calls and waiting times.

This enables subsequent tasks to proceed quickly and optimizes end-to-end latency.

**Tail Latency and Fairness.** The advantages of HeraSys in reducing tail latency and enhancing fairness for long queries are shown in the P95/P99 columns of Table 1. Due to the lack of application-level information, LangGraph and LlamaIndex experience severe HOL Blocking once they encounter long document processing or generation tasks, causing P95 and P99 latency to increase sharply under high concurrency. Even Ayo, which supports fine-grained parallelism, maintains high tail latency under high load due to the lack of guarantees for long-tail tasks, making it difficult to meet strict SLO requirements. In contrast, HeraSys reduces P95 latency by 48.5% and P99 latency by 53.9%. This confirms the effectiveness of our Resource Skewing mechanism, which successfully avoids the starvation issues common in pure SRTF strategies by enforcing a minimum resource reservation for long-tail tasks. Consequently, HeraSys pro-

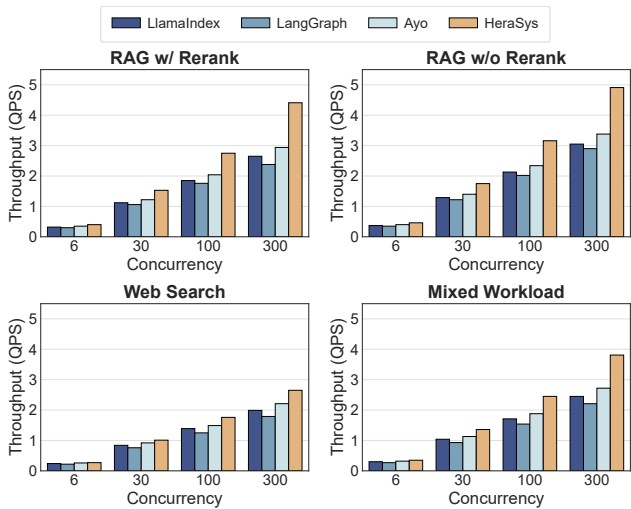

*Figure 6.* Evaluation of service throughput under different concurrency settings and workloads.

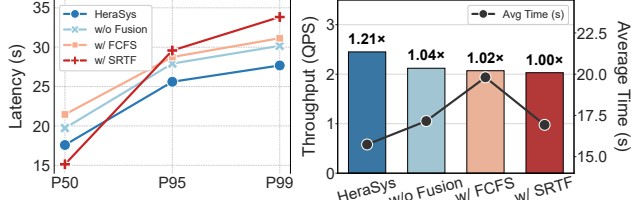

*Figure 7.* Ablation study of HeraSys core components under the mixed workload with 100 concurrency. Note that "w/ FCFS" and "w/ SRTF" represent the system variants using FCFS and SRTF scheduling strategies.

*Table 2.* Variations in the Average Number of Nodes and Latency with Adaptive Pipeline Decomposition on RAG w/ Rerank. Note that Ayo and "HeraSys (w/o)" employ a fixed decomposition strategy, statically splitting the workflow into two parallel pipelines, whereas HeraSys adjusts the pipeline granularity based on load.

| Load | Config | Nodes | Latency (s) |
|---|---|---|---|
| Low Concurrency 6 | Ayo | 19.0 | 3.71 |
| | HeraSys (w/o) | 19.0 | 3.56 |
| | **HeraSys** | **24.7** | **3.48** |
| High Concurrency 100 | Ayo | 19.0 | 19.09 |
| | HeraSys (w/o) | 19.0 | 17.13 |
| | **HeraSys** | **15.6** | **15.77** |

vides a fairer and more predictable quality of service while ensuring the response speed of short tasks.

### 4.3. Throughput Scalability

We further measure the service throughput (QPS) of each system under latency constraints, as shown in Figure 6. As concurrent pressure increases, HeraSys shows higher throughput scalability, achieving a throughput up to 1.50 times that of Ayo, and nearly 2 times that of traditional modular frameworks (LangGraph and LlamaIndex). This significant performance gap highlights the limitations of baseline systems: both LangGraph and Ayo essentially treat each request as an independent computation graph. Even when multiple users simultaneously query hot documents, these systems repeatedly execute the same Embedding and retrieval operators, resulting in wasted resources. Conversely, HeraSys's throughput gains result from the deduplication capabilities of the Graph Fuser. As concurrency increases, the probability of sharing subgraphs and nodes between requests increases, allowing HeraSys to identify and merge these redundant computations. Simultaneously, HeraSys's load-adaptive regulation maintains higher batch sizes and reduced pipeline decomposition granularity under high load, thereby reducing scheduling overhead. This allows the system to convert increased load pressure into efficient deduplication and adaptive regulation, supporting higher service throughput than baselines on the same physical resources.

### 4.4. Mechanism Contribution and Ablation

To quantify the contribution of each core mechanism of HeraSys, we conduct an ablation study, as shown in Figure 7 and Table 2.

**Graph Fusion and Scheduling Strategy.** Under the mixed workload with high concurrency (Concurrency 100), we conduct tests on the Graph Fuser and Runtime Scheduler to assess their impacts on system performance. Figure 7 shows that disabling the Graph Fuser (w/o Fusion) limits system capacity; the total throughput of Full HeraSys is 1.16× that without fusion. This confirms that scheduling optimization alone is insufficient, and redundancy removal is crucial for improving throughput. In terms of latency, reverting the Load-Aware Scheduler to a standard FCFS strategy increases P99 latency by 12.5%. While using only the SRTF strategy reduces P50 (by approximately 13%), it increases P99 latency by 22.3% compared to Full HeraSys. This highlights the limitations of single strategies in complex workflows and the value of HeraSys's joint optimization, which balances average speed and long-tail fairness.

**Adaptive Pipeline Decomposition.** We perform comparative experiments on RAG w/ Rerank workflows under varying load conditions to further verify the effectiveness of adaptive pipeline decomposition. Table 2 shows how adaptive pipeline decomposition dynamically adjusts graph granularity based on load. Under low load (the first section of Table 2), HeraSys increases the average number of nodes per query from 19.0 to 24.7 (+30%), performing finer-grained splitting to utilize idle resources for parallel computation, thereby reducing latency. Conversely, under high load (the second section of Table 2), HeraSys reduces the average node count to 15.6 (-18%) to reduce overhead. HeraSys does

*Table 3.* Parameter sensitivity under the mixed workload with 100 concurrency.

| Config | P50 (s) | P95 (s) | P99 (s) | Avg (s) | QPS |
|---|---|---|---|---|---|
| Default | 17.58 | 25.61 | 27.68 | 15.74 | 2.45 |
| Short-first | **16.54** | 26.24 | 28.73 | 15.89 | 2.39 |
| Throughput | 17.94 | 25.87 | 28.11 | **15.43** | **2.60** |
| Tail-fairness | 18.29 | **25.18** | **27.06** | 16.31 | 2.33 |

not rely on static configuration but balances parallelism and overhead, ensuring high performance across varying loads.

### 4.5. Parameter Sensitivity

We evaluate the sensitivity of HeraSys to scheduler parameters under the mixed workload with 100 concurrency. We keep the system setting unchanged and vary only the parameters in the joint priority function and resource skewing. As shown in Table 3, different coarse configurations lead to different performance tendencies: "Short-first" shifts the system toward lower median latency, "Throughput" shifts it toward higher average efficiency and QPS, and "Tail-fairness" shifts it toward lower high-percentile latency. This reflects the role of these parameters in balancing short-query responsiveness, structural efficiency, and long-query protection in mixed serving scenarios.

## 5. Conclusion

In this paper, we presented HeraSys, a fine-grained end-to-end serving system designed for multi-workflow scenarios, enabling cross-workflow optimization. Building upon this architecture, we designed a dual-layer optimization mechanism: prior to execution, the system identified and consolidated cross-query computational redundancy, reducing system overhead; at runtime, HeraSys introduced a joint scheduling strategy that dynamically evaluated task characteristics and system load to adjust execution order and data granularity, effectively addressing resource contention in heterogeneous workloads. We evaluated HeraSys through extensive experiments across diverse scenarios, which validated the effectiveness of the proposed architecture. We believe that the cross-workflow optimization approach in HeraSys provides insights for building efficient and scalable LLM application serving systems.

## Acknowledgments

This work was supported in part by the National Natural Science Foundation of China (NSFC) under Grant 62302048, Grant U25A20436, and Grant 62272050; in part by Guangdong Higher Education Association under Grant 24GQN97; in part by the Guangdong Provincial Higher Education Institutions under Grant 2024KTSCX219; and in part by Beijing Normal University at Zhuhai Education Reform Project under Grant jx2025037.

## Impact Statement

This paper presents work whose goal is to advance the field of Machine Learning systems by optimizing the serving efficiency of Large Language Models. This work contributes to reducing computational resource consumption and operational costs, thereby fostering the development of greener and more efficient LLM serving. Additionally, it significantly enhances the user experience of LLM applications by ensuring faster and fairer responses. We do not foresee any direct negative societal consequences, as this research focuses on the underlying serving infrastructure rather than generative content capabilities.

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

# A. Problem Modeling and Solving Strategy

This appendix provides the mathematical modeling, constraint definitions, and design of the online solving strategy for HeraSys.

## A.1. Problem Formulation

### A.1.1. WORKLOAD AND GRAPH MODEL

Let $\mathcal{Q}_t$ be the set of active queries at time $t$ (i.e., queries that have arrived by $t$ and are not yet completed). Each query $q \in \mathcal{Q}_t$ is represented by a Directed Acyclic Graph (DAG) $G_q = (\mathcal{V}_q, \mathcal{E}_q)$, where nodes represent operators or sub-tasks, and edges represent execution dependencies.

Each node $v \in \mathcal{V}_q$ has the following attributes:

1. **Workload** ($w_v$)**:** Estimated computation volume. For non-LLM nodes, this represents input text size or FLOPs; for LLM nodes, it represents the expected number of tokens.

2. **Compatible Resources** ($p_v$)**:** A set of executable engine/device types (e.g., Embedding instances, LLM instances).

3. **Reverse Topological Depth** ($d_v$)**:** The number of layers from the node to the sink node. Smaller values indicate proximity to the final output.

4. **Splittability** ($s_v$)**:** A flag and upper bound indicating if the node allows parallel splitting into up to $s_v$ execution units. If node $v$ is split with factor $k_v > 1$, it is executed as $\{v_i\}_{i=1}^{k_v}$ in parallel, and the logical node completes when all units complete: $C_v = \max_i C_{v,i}$.

5. **Performance Profile** ($\theta_v$)**:** Labels for operator type, overhead, and parallel efficiency, used for throughput function calibration.

6. **Overhead** ($overhead_v$)**:** Fixed scheduling and synchronization overhead (e.g., kernel launch, split/merge, communication), used in the runtime model.

7. **Node Contribution** ($\phi_v$)**:** A metric quantifying the contribution of node $v$ to query progress. It integrates three topological features: downstream remaining workload, downstream path length, and the number of enabled parallel branches. Defined as:
$$\phi_v = \sum_{u \in \mathcal{N}_v} (\omega_1 \cdot w_u + \omega_2 \cdot d_u) + \omega_3 \cdot |\mathcal{N}_v|, \tag{9}$$

where $\mathcal{N}_v$ is the set of successor nodes of $v$. We determine weights $\omega$ based on dimensional normalization and prior empirical experience. Nodes with high $\phi_v$ are prioritized to accelerate the current workflow.

### A.1.2. RESOURCE MODEL

Let $\mathcal{R}$ be **the set of resource types**. Each resource type $r \in \mathcal{R}$ is characterized by its parallel capacity $cap_r$ (the number of parallel engines/instances) and batching limit $B_r$.

### A.1.3. DECISION VARIABLES

At a scheduling event $t$ (subscript omitted for brevity), let $\mathcal{V}_{ready}(t)$ denote the set of ready nodes whose predecessors are completed. The system determines (for $v \in \mathcal{V}_{ready}(t)$):

1. $x_{v,r} \in \{0, 1\}$: Whether node $v$ is scheduled on resource type $r$ at this event (unscheduled nodes have $x_{v,r} = 0$ for all $r$).

2. $b_{v,r} \in \mathbb{N}^+$: The **batch size** formed for node $v$ on resource $r$ (constrained by $B_r$).

3. $a_{v,r} \in [0, 1]$: **Resource share allocation** (e.g., GPU memory/SM ratio, CPU cores) on type $r$ for node $v$.

4. $k_v \in \mathbb{N}^+$: **Split factor** (number of parallel execution units) for node $v$, where $k_v = 1$ means no splitting.

5. $F_v$: Start time of node $v$.

6. $T_v$: Execution duration of node $v$.

7. $C_v$: Completion time of node $v$, where $C_v = F_v + T_v$.

We use $q(v)$ to denote the query that contains node $v$. The completion time of query $q$ is $C_q = \max_{v \in \mathcal{V}_q} C_v$. If $k_v > 1$, we denote the $i$-th execution unit by $v_i$ with auxiliary runtime $T_{v,i}$ and completion time $C_{v,i} = F_v + T_{v,i}$, and the logical node completion time is $C_v = \max_i C_{v,i}$.

### A.1.4. AUXILIARY FUNCTIONS

**Throughput Function:** $\mu_r(b_{v,r}, a_{v,r}; \theta_v)$ denotes the throughput rate on resource $r$ with batch size $b$ and share $a$, calibrated by node profile $\theta_v$.

**Remaining Service Time Estimation:** The estimated remaining time for query $q$, $\widehat{S}_q$, is defined as:

$$\widehat{S}_q(t) = \mathcal{F}\left(\frac{A_q(t)}{\sum_{v \in \mathcal{V}^q_{done}(t)} \bar{T}_v + \epsilon_T}\right) \cdot \sum_{v \in \mathcal{V}^q_{remain}(t)} \bar{T}_v, \tag{10}$$

where $\mathcal{V}^q_{done}(t)$ and $\mathcal{V}^q_{remain}(t)$ denote the completed and remaining nodes of query $q$ at time $t$, $\bar{T}_v$ denotes the predicted runtime of node $v$, and $A_q(t)$ is the attained (accumulated) service time of $q$ so far (sum of measured runtimes of completed (sub-)nodes). $\epsilon_T > 0$ avoids division by zero at the beginning. $\mathcal{F}$ is a runtime correction factor (e.g., if LLM inference exceeds expectations, $\mathcal{F}$ increases $\widehat{S}_q$). **PLAS Heuristic:** (Luo et al., 2025) Since LLM decoding is unpredictable and often non-preemptive, we decompose a decoding node into sequential sub-nodes (decoding-1, decoding-2, ...), creating preemption points. After each sub-node, we update $A_q(t)$ with the observed runtime and refresh $\widehat{S}_q(t)$.

### A.1.5. OBJECTIVE FUNCTION

Our objective is to balance average latency with fairness for long queries. We define the objective function as:

$$\min \quad \mathcal{J} = \lambda \cdot \frac{1}{|\mathcal{Q}_t|} \sum_{q \in \mathcal{Q}_t} C_q + (1 - \lambda) \cdot \max_{q \in \mathcal{Q}_t} C_q, \tag{11}$$

where the first term represents average completion time (optimizing throughput/short queries), and the second term represents maximum completion time (optimizing tail latency). In the rolling-horizon setting, $C_q$ represents the completion time under the current commitments plus the candidate decision at this event. $\lambda \in [0, 1]$ is an adaptive balancing coefficient. $\lambda$ varies based on system load, depending on the number of active queries and pending nodes.

### A.2. Constraints

#### A.2.1. HARD CONSTRAINTS

1. **Dependency Order:** For each query $q \in \mathcal{Q}_t$ and each node $v \in \mathcal{V}_q$,

$$F_v \geq \max_{\{u:u \rightarrow v \in \mathcal{E}_q\}} C_u. \tag{12}$$

2. **Compatibility, Capacity & Exclusivity:**

   - **Compatibility:** $x_{v,r} = 0, \forall r \notin p_v$.
   - **Capacity:** $\sum_v a_{v,r} \leq cap_r, \forall r \in \mathcal{R}$.
   - **Exclusivity (allow waiting):** $\sum_r x_{v,r} \leq 1$ and $0 \leq a_{v,r} \leq x_{v,r}, \forall v$. **No hybrid execution:** a node (if scheduled) runs on a single resource type.

3. **Batching & Throughput:**

   - If $x_{v,r} = 1$: $1 \leq b_{v,r} \leq B_r$. Batching occurs only among ready nodes with consistent $\theta_v$ and parallelizability.

- If $x_{v,r} = 1$, we define the per-unit runtime

$$T_{v,i} = overhead_v + \frac{w_{v,i}}{\mu_r\left(b_{v,r}, \frac{a_{v,r}}{k_v}; \theta_v\right)}, \quad i = 1, \ldots, k_v, \tag{13}$$

and the node runtime/finish time as

$$T_v = \max_{i \in \{1, \ldots, k_v\}} T_{v,i}, \quad C_v = F_v + T_v. \tag{14}$$

4. **Parallel Splitting & Load Balancing:**

- $1 \leq k_v \leq s_v, \forall v \in \mathcal{V}_q$.
- If $k_v > 1$, we execute $v$ as $\{v_i\}_{i=1}^{k_v}$ with an equal workload split:

$$w_{v,i} = \frac{w_v}{k_v}, \quad i = 1, \ldots, k_v, \tag{15}$$

and assume equal resource-share split across units, i.e., each unit receives $a_{v,r}/k_v$ on the chosen resource type. The logical node completes when all units complete, i.e., $C_v = \max_i C_{v,i}$. We include synchronization overheads in $overhead_v$.

- **Pipeline split consistency:** For a decomposed linear chain, we enforce the same split factor along pipeline edges $\mathcal{E}_q^{pipe} \subseteq \mathcal{E}_q$:

$$k_u = k_v, \quad \forall (u \to v) \in \mathcal{E}_q^{pipe} \subseteq \mathcal{E}_q. \tag{16}$$

A.2.2. SCHEDULING STRATEGIES (OPTIONAL SOFT CONSTRAINT FORMULATION)

We model these strategies as soft constraints with penalties, though implementations may use scoring logic. Here $\varepsilon_\phi > 0$ and $\varepsilon_S > 0$ are penalty weights.

1. **Downstream Contribution Priority:** For competing ready nodes $u, v$ on the same resource type, if $\phi_u > \phi_v$, then $u$ enables more downstream work. We encourage $u$ to start no later than $v$ via a hinge penalty:

$$\varepsilon_\phi \cdot \sum_{(u,v):\phi_u > \phi_v} \max\{0, F_u - F_v\}. \tag{17}$$

2. **Short Query Priority (SRTF):** For competing ready nodes $v \in \mathcal{V}_q, v' \in \mathcal{V}_{q'}$ on the same resource type, if $\widehat{S}_q < \widehat{S}_{q'}$, we apply a penalty for scheduling the short query behind the long one:

$$\varepsilon_S \cdot \max\{0, F_v - F_{v'}\}. \tag{18}$$

3. **Long Query Resource Skewing:** We monitor the distribution $(\mu_s, \sigma_s)$ (mean and standard deviation) of active $\widehat{S}_q$. We define a dynamic threshold $\tau_{load} = \mu_s + \eta \cdot \sigma_s$, where $\eta > 0$ is a hyperparameter.

- If $\widehat{S}_q > \tau_{load}$, the system marks query $q$ as a long query. Its heavy nodes are marked $y_v = 1$ (otherwise $y_v = 0$).
- **Flexible resource reservation:** For each resource type $r \in \mathcal{R}$, define the set of ready heavy nodes $\mathcal{H}_r(t) = \{v \in \mathcal{V}_{ready}(t) : y_v = 1, r \in p_v\}$. If $\mathcal{H}_r(t) \neq \emptyset$, we reserve a minimum share for heavy nodes:

$$\sum_{v \in \mathcal{H}_r(t)} a_{v,r} \geq \rho \cdot cap_r, \quad \forall r \in \mathcal{R}, \tag{19}$$

where $\rho \in (0, 1]$ is the reserved fraction. This reservation is flexible: if $\mathcal{H}_r(t) = \emptyset$, the system allocates it to any ready nodes; when heavy nodes exist, we serve them with higher priority within the reserved budget.

- Heavy nodes receive minimum resource guarantees and higher splitting: if $x_{v,r} = 1$, we enforce $a_{v,r} \geq \eta_{long} \cdot y_v$ and $k_v \geq \kappa_{long} \cdot y_v$, where $\eta_{long} \in (0, 1]$ and $\kappa_{long} \in \mathbb{N}^+$ are hyperparameters.

4. **Redundancy Elimination & Result Sharing:** We merge nodes $\mathcal{M}(v^*)$ identified as semantically identical into a single execution $v^*$.

- Nodes $v \in \mathcal{M}(v^*)$ depend on the output of $v^*$.
- The contribution of the representative node is the sum: $\phi_{v^*} = \sum_{v \in \mathcal{M}(v^*)} \phi_v$. This naturally increases the priority of shared nodes.

## A.3. Online Solving Strategy

### A.3.1. ONLINE FRAMEWORK

1. **Rolling Horizon + Greedy/Approximation:** At each event $t$, we make decisions only for ready nodes of active queries in $\mathcal{Q}_t$. We solve the optimization problem (with soft constraints) for the ready set, update states $(x, b, a, k, F, T, C)$, and advance to the next event.

2. **$\lambda$-Adaptation:** The balancing coefficient $\lambda$ adapts to load. Under high load, we increase $\lambda$ to lower average completion time; when tail latency increases, we decrease $\lambda$ to suppress $C_{max}$.

3. **Non-Static Critical Path:** We drive scheduling using SRTF/PLAS heuristics and Downstream Contribution ($\phi_v$), using resource skewing to handle long tails, rather than static critical path analysis.

### A.3.2. ALGORITHM DESIGN

The execution logic of the scheduling is formalized in Algorithm 1, illustrating how the online scheduler performs decisions upon being triggered by events, including New Query Arrivals, Node Completions, and Resource Releases.

---

**Algorithm 1** Pseudocode of the online scheduler.

---

**Require:** Active queries $\mathcal{Q}_t$; resource types $\mathcal{R}$ with $cap_r$ and $B_r$; parameters $(\alpha, \gamma, \delta, \rho)$.
1: **On each event** $e \in \{arrival, completion, release\}$.
2: ▷ *arrival*: new query arrives; *completion*: a node finishes; *release*: a resource becomes available.
3: **Phase 1: State Update & Ready Set Construction**
4: **for all** $q \in \mathcal{Q}_t$ **do**
5:     Update attained service time $A_q(t)$ and remaining-time estimate $\widehat{S}_q(t)$.
6: **end for**
7: Construct ready set $\mathcal{V}_{ready}(t) = \{v : \text{ all predecessors of } v \text{ are completed}\}$.
8: Merge semantic duplicates $\mathcal{M}(v^*)$ into representatives $v^*$ for reuse and result sharing.
9: Insert each ready node into per-resource queues ($short\_queue_r$ / $long\_queue_r$) for all compatible $r \in p_v$.
10: **Phase 2: Prioritization**
11: **for all** $v \in \mathcal{V}_{ready}(t)$ **do**
12:     Compute contribution $\phi_v$ and priority $\pi_v = \alpha/(\widehat{S}_{q(v)} + \epsilon) + \gamma\phi_v + \delta y_v$.
13:     Update the priority key of $v$ in the corresponding queue(s).
14: **end for**
15: **Phase 3: Joint Allocation**
16: **for all** $r \in \mathcal{R}$ **do**
17:     **while** $cap_r$ remains and queues are non-empty **do**
18:         Select from $short\_queue_r$ and $long\_queue_r$ (skewing/reservation with $\rho$ when applicable).
19:         Form candidate batch $Batch$ from queue head (consistent type, limit $B_r$).
20:         Search feasible $(b, a)$; compute $T_v$ for $v \in Batch$; pick the configuration with max net gain.
21:         For heavy nodes ($y_v = 1, s_v > 1$), choose split $k_v \in [1, s_v]$ based on $\lambda$.
22:         Commit $(x, b, a, k, F, T, C)$, update resource availability and $\widehat{S}_q$, and proceed to next event.
23:     **end while**
24: **end for**
25: **return** $C_q = \max C_v$ when all queries are complete.

---

**Phase 1: State Update & Ready Set Construction**

1. Online update of $A_q(t)$ and $\widehat{S}_q(t)$ for SRTF/PLAS metrics.

2. Form **Ready Set**: Nodes with all predecessors completed. Specifically, for each resource type $r$, we maintain two queues ($short\_queue_r$ and $long\_queue_r$). We insert a ready node $v$ into queues of its compatible types $r \in p_v$, and route it to $short\_queue_r$ or $long\_queue_r$ based on its query label.

3. **Redundancy Detection:** We merge semantically matching nodes $\mathcal{M}(v^*)$ into representative node $v^*$ to reduce computation and latency.

**Phase 2: Prioritization** Calculate priority score $\pi_v$ for each ready node:

$$\pi_v = \frac{\alpha}{\widehat{S}_{q(v)} + \epsilon} + \gamma \cdot \phi_v + \delta \cdot y_v, \tag{20}$$

where $\epsilon > 0$ prevents division by zero.

- **Term 1 (SRTF/PLAS):** Prioritizes queries with small remaining time $\widehat{S}_q$; $\widehat{S}_q$ is updated online after decoding sub-nodes using PLAS-inspired correction.

- **Term 2 (Topology):** Prioritizes critical nodes enabling downstream work ($\phi_v$).

- **Term 3 (Anti-Starvation):** Increases priority for heavy nodes ($y_v$) in long queries.

We maintain each *short_queue$_r$* and *long_queue$_r$* as a priority queue keyed by $\pi_v$ (descending).

**Phase 3: Joint Allocation** For each resource type $r$, while capacity/shares remain:

1. **Batch Construction:** We select the next node from *short_queue$_r$* and *long_queue$_r$* (e.g., pick the higher $\pi_v$), then extract a candidate batch ($Batch$) from the queue head (consistent type, limit $B_r$).

2. **Determine Batch Size ($b$) & Share ($a$):**
   - We search the set $b \in \{1, \ldots, \min(B_r, |Batch|)\}$ and candidate strategies for $a$ (Equal/Skewed/Weighted).
   - We compute $T_v$ for $\forall v \in Batch$.
   - We calculate the score based on marginal improvement to $\mathcal{J}$ (approx.) minus soft constraint penalties.
   - We select the configuration with maximum net gain.

3. **Parallel Splitting ($k$):** For heavy nodes ($y_v = 1, s_v > 1$), we split with $k_v \in [1, s_v]$ based on $\lambda$. We use equal workload split $w_{v,i} = w_v/k_v$ and treat $\{v_i\}_{i=1}^{k_v}$ as parallel execution units sharing the batch configuration.

4. **Enforcing Skewing:** If $y_v = 1$ and $x_{v,r} = 1$, we ensure $a_{v,r}$ meets $\eta_{long}$.

5. **Commit:** We fix $(x, b, a, k, F, T, C)$, update resource availability and $\widehat{S}_q$, and proceed to next event.

**Termination:** When all queries are complete, output $C_q = \max C_v$.

## B. Theoretical Analysis

### B.1. NP-Hardness Proof

**Theorem:** The (offline) HeraSys scheduling problem is NP-hard.

**Proof:** We reduce from the precedence constrained scheduling on identical parallel machines, $P|prec|C_{max}$, which is known to be NP-hard. Consider an instance of $P|prec|C_{max}$ with $n$ jobs, processing times $\{p_j\}$, precedence constraints, and $m$ identical machines. We construct a HeraSys instance as follows:

- Create a single query $q$ whose DAG nodes $\{v_j\}_{j=1}^n$ correspond to jobs, and add edges to match the precedence constraints.

- Use a single resource type $r$ with capacity $m$, disable batching ($B_r = 1$), and disable splitting ($s_{v_j} = 1$ for all $j$).

- Set $p_{v_j} = \{r\}$, $overhead_{v_j} = 0$, and choose $\theta_{v_j}$ such that $\mu_r(1, 1; \theta_{v_j}) = 1$, yielding $T_{v_j} = w_{v_j} = p_j$ when scheduled.

Since the active query set size $|\mathcal{Q}_t| = 1$, the objective reduces to minimizing $C_q = \max_{v \in \mathcal{V}_q} C_v$, which equals the makespan $C_{max}$. Any feasible schedule for the HeraSys instance corresponds to a feasible schedule for the original $P|prec|C_{max}$ instance with the same makespan, and vice versa. Therefore, an optimal solution to HeraSys yields an optimal solution to $P|prec|C_{max}$, implying HeraSys is NP-hard.

## B.2. Algorithm Properties and Advantages

### B.2.1. COMPETITIVENESS AND OPTIMALITY

Since future arrival times are unknown, this is an online algorithm.

- **SRTF Time Priority (with PLAS correction):** The first term of $\pi_v$ $(1/(\widehat{S}_q + \epsilon))$ approximates the SRTF policy. Theoretically, SRTF minimizes the average waiting time in single-processor preemptive scheduling environments (Schrage, 1968). In our setting, we update $\widehat{S}_q$ online at decoding sub-node boundaries using attained service time $A_q(t)$, based on PLAS.

- **Preemption Design & Decoding Decomposition:** In traditional LLM serving, the unpredictability of decoding tasks often leads to non-preemptive behavior. We decompose decoding operators into sequentially dependent sub-nodes. Upon completion of a sub-node, we update $A_q(t)$ with the observed runtime and refresh $\widehat{S}_q(t)$, enabling PLAS-style priority decay for long-attained queries while preserving preemption points.

- **Sub-optimality:** Due to the greedy strategy (considering only current Ready nodes), global optimality is not guaranteed. For example, executing a short task now might block a soon-to-arrive critical long task. The $\lambda$-adaptation and resource skewing mechanisms attempt to balance average and worst-case performance.

### B.2.2. LIVENESS AND SAFETY

- **Deadlock-Free:** The DAG structure guarantees acyclic dependencies. The Ready Set contains only nodes with completed predecessors. Resource allocation is atomic per event, preventing hold-and-wait cycles.

- **Starvation Mitigation (Heuristic):** The algorithm prevents starving long queries via resource reservation, heavy-node boosting ($y_v$), minimum share $\eta_{long}$, and adaptive $\lambda$. A formal starvation-free guarantee can be obtained by adding an *aging* term to $\pi_v$ (or enforcing a hard priority floor once a query waits longer than a threshold), which we leave as an optional extension.

### B.2.3. RATIONALE AGAINST STATIC CRITICAL PATH

In online scheduling scenarios constrained by limited and non-fully parallel resources, determining the "Critical Path" is computationally infeasible at the moment of decision-making. The actual critical path typically emerges only post-facto after scheduling is complete, rather than serving as available a priori information (accurately predicting the critical path beforehand incurs significant computational overhead and costs).

- $F_v$ depends on contention, batching, splitting, and future arrivals, which are unknown a priori.

- Forcing priority for the estimated longest path can degrade $\sum C_q$. *Counter-example:* A long query $q_L$ with a long path but high parallelism might block critical resources needed by multiple short, resource-sensitive queries $q_S$.

- Instead, we use Node Topology Contribution ($\phi_v$) as a proxy for Longest Remaining Path adapted for dynamic environments, identifying potential bottlenecks in real-time.

## B.3. Complexity Analysis

We analyze the computational complexity of the HeraSys online scheduling algorithm for a single decision event. Let $N$ denote the total number of active nodes in the system, and $N_{ready}$ denote the number of ready nodes (subset of $N$ whose dependencies are satisfied) at the current timestamp. Let $|\mathcal{R}|$ be the number of distinct resource types. We introduce $B_{max} = \max_r B_r$ as the maximum batch size limit and $S_{max} = \max_v s_v$ as the maximum split factor.

The scheduling procedure is divided into three sequential phases. We analyze the asymptotic complexity for each:

**1. Phase 1: State Update and Redundancy Elimination**

- **State Update:** Updating the remaining service time $\widehat{S}_q$ and identifying long queries involves iterating through active queries. Since node states update incrementally upon completion events, the overhead is linear with respect to the number of nodes processed in the current event window, bounded by $O(N_{ready})$.

- **Redundancy Elimination:** The system employs a hash-based mechanism to detect duplicates. For each ready node, computing the semantic hash and querying the global duplication map takes $O(1)$ time on average (assuming a good hash function). Thus, processing the entire ready set takes $O(N_{ready})$.

**2. Phase 2: Prioritization and Sorting**

- **Scoring:** Calculating the priority score $\pi_v$ for all ready nodes involves constant-time arithmetic operations per node, totaling $O(N_{ready})$.

- **Sorting:** The ready nodes are sorted to form candidate queues for each resource type. Using an efficient comparison-based sorting algorithm (e.g., Quicksort or Timsort), this operation requires $O(N_{ready} \log N_{ready})$. This step typically dominates the complexity of this phase.

**3. Phase 3: Joint Allocation** The allocator iterates through resource types and greedily assigns tasks.

- **Search Space:** For each allocation decision, the algorithm constructs a candidate batch and searches for the optimal configuration $(b, a, k)$. The search iterates through feasible batch sizes $b \in [1, \min(|Batch|, B_{max})]$ and split counts $k \in [1, S_{max}]$.

- **Cost per Decision:** The configuration search costs $O(B_{max} \cdot S_{max})$. Scoring each configuration uses a local surrogate of $\Delta\mathcal{J}$ computed from cached per-query estimates (thus $O(1)$ per configuration). If one instead recomputes the exact global objective with full dependency propagation, the worst-case cost per configuration can be $O(|\mathcal{E}|)$.

- **Total Allocation Cost:** In the worst case, the allocator iterates through all nodes in the candidate queues once to assign them or determine they cannot be scheduled, resulting in $O(|\mathcal{R}| \cdot N_{ready} \cdot B_{max} \cdot S_{max})$, which is linear in $N_{ready}$ when $B_{max}, S_{max}$ are treated as constants.

**Overall Complexity** Aggregating the three phases, the total time complexity per scheduling event is dominated by the sorting operation in Phase 2 (under the local-surrogate scoring implementation):

$$\mathcal{T}_{event} = O(N_{ready} \log N_{ready}) \tag{21}$$

Note that $N_{ready}$ is constrained by DAG dependencies and represents only the **execution frontier** (i.e., the instantaneous available parallelism) of the graph, satisfying $N_{ready} \ll N$. Given that LLM inference tasks (especially the prefill and decoding phases) are computationally intensive, the scheduling overhead incurred by this log-linear complexity is negligible relative to task execution time. This low asymptotic complexity ensures that HeraSys scales efficiently to high-concurrency scenarios without becoming a system bottleneck.

**B.4. Limitations**

1. **Dependency on $\widehat{S}_q$ Estimation:** The algorithm relies on $\widehat{S}_q$ for SRTF. While decomposition assists correction, large prediction errors (common in LLM generation) can degrade performance.

2. **Local Minima:** Greedy batching might force high-priority nodes to wait to fill a batch, or optimize throughput at the expense of $\max C_q$. $\lambda$-adaptation mitigates this but does not eliminate it.

3. **Splitting Overhead:** Splitting reduces latency but introduces synchronization overhead ($overhead_v$). If split too finely, Amdahl's Law limits speedup, and costs may outweigh benefits.

