# OpenReview forum: "HeraSys: Collaborative Serving of Multiple LLM Workflows via Fine-Grained End-to-End Optimization"
_ICML.cc/2026/Conference — ICML 2026 regular_

### Official Review · Reviewer_6qaw · 2026-02-21

**Soundness:** 2
**Presentation:** 3
**Significance:** 2
**Originality:** 2
**Overall Recommendation:** 4
**Confidence:** 4

**Summary:**

Existing LLM serving systems struggle with multi-tenant agentic workflows because they process requests in isolation, ignoring cross-workflow redundancies and causing resource starvation for long tasks. To resolve this, the authors introduce HeraSys, an end-to-end system utilizing a Pre-Execution Graph Fuser that identifies and merges structurally identical subgraphs across concurrent queries to eliminate computational waste. HeraSys also features a Runtime Load-Aware Joint Scheduler that balances overall efficiency and fairness by combining Shortest Remaining Time First (SRTF) heuristics with strict resource reservations for long-running queries. Extensive evaluations show that the proposed framework significantly reduce tail latency and improve system throughput.

**Compliance With Llm Reviewing Policy:**

Affirmed.

**Final Justification:**

all my concerns are addressed

**Key Questions For Authors:**

Thank you for submitting the work to ICML. I have the follow questions for the authors:

* In the introduction, the authors mention that “multiple concurrent queries often involve embedding and retrieving the same batch of common document chunks”. Do the authors have any data, traces, or literature to back up this claim? Since a significant part of the paper's core idea is built on the premise that concurrent queries frequently fetch common document embeddings, providing such data and citations would make the argument much more convincing.
* Regarding Figure 1c and the dynamic fusion and reuse mechanism in Section 3.3: Can the authors explain why a simple prefix-caching mechanism in existing serving frameworks (e.g., SGLang or vLLM) would not resolve this issue? For two concurrent nodes that share the same prefix (such as tool calling), unless they are executed in the exact same batch, only one of them will have its cache computed, while the other will simply utilize prefix caching, correct? It would be highly beneficial for the authors to explicitly clarify the performance gap between standard prefix caching and their proposed mechanism.
* The formulation of the remaining job time is questionable. First, how does the system handle early nodes in the graph, given that the attained time is very short and the ratio of attained time to the expected runtime of the finished nodes might not be reliable? How does the corrector function $\mathcal{F}$ help in this specific scenario? Second (and this is perhaps the most significant issue with the formulation), the current approach uses a proportional heuristic. For example, if the job has been running $1.5\times$ slower than expected so far, it assumes the rest of the job will also run $1.5\times$ slower, adjusted by the corrector $\mathcal{F}$. However, nodes may have vastly different resource demands; some tasks might be compute-intensive (e.g., a long prefill), while others might be memory-bandwidth intensive (e.g., a long decode). Simply applying a uniform proportional heuristic without considering the specific hardware characteristics of the task running on each node seems flawed.
* Can the authors clarify what they mean by “compute node semantic hashes via SHA-256”? What exactly is a semantic hash in this context? Does it imply semantic encoding? SHA-256 is strictly a cryptographic hash function, and it does not perform any form of semantic encoding.
* The evaluation results are quite difficult to interpret regarding the load. What exactly do the authors mean by "concurrency" here? What do the numbers 6, 30, 100, and 300 refer to? Do they represent the incoming request rate, or the total number of requests being served simultaneously within the system?
* Can the author provide more clarity on how they tune the hyperparameters in the system. Considering the proposed system has lots of hyperparameters, this feels need to be done in a systematic way.

**Limitations:**

Same as the questions asked.

**Strengths And Weaknesses:**

**Strengths**
* The paper addresses a timely topic.
* The proposed system is sound. The idea of partitioning the DAG and fusing nodes before sending them to the LLM engine is an interesting approach.
* The framework proposes a load-aware joint scheduling mechanism with adaptive execution adjustments.
* The evaluation setup—including the baselines and workloads—covers a wide and comprehensive range of traffic and request patterns.

**Weaknesses**
* The remaining job running time estimation formulation is not properly evaluated; it requires an ablation study to show that this component actually works as expected. Additionally, the corrector function $\mathcal{F}$ should be clearly specified (I assume this is a smoothing function to mitigate the influence of early nodes?)
* The joint priority scheduling formula seems contradictory. First, what exactly is $\hat{S}_{q(v)}$? Does it represent the remaining time of a query $q$ given that the query is executing node $v$ now? If so, for an early heavy node in a critical path, the remaining time estimation would be long, which reduces the priority, but the node contribution would be high due to the number of successors and downstream workload. Moreoever, this priority calculation appears highly dependent on hyperparameter tuning, which is not evaluated in the experiments section.
* Based on the descriptions of the node semantic signature in Section 3.3 and Section 4.1, I believe "semantic" is not the appropriate term. It seems that only a simple hash is performed on the node to check for exact equivalence; no actual semantic analysis or embedding is involved in the process. Please clarify if I am mistaken.

---

> ### Author Rebuttal · Authors · 2026-03-31
>
> We thank the reviewer for the careful reading, especially on shared chunks, prefix caching, and remaining-time estimation.
>
> **Q1.**
> - **There is prior evidence that shared document chunks and cross-request overlap are common in realistic LLM workloads.** Our evaluation does not claim one universal exact duplication rate; it reflects an overlap pattern. Long contexts, templates, tool instructions, and retrieved chunks overlap. Preble (ICLR 2025) studies five workloads and reports prompt-to-output ratios of 37x-2494x, 85%-97% shared prompt tokens, and 8.6-126 average reuses of shared sequences. Prompt Cache (MLSys 2024) identifies system messages, prompt templates, and context documents as common overlaps. CacheBlend (EuroSys 2025) and IMPRESS (FAST 2025) show reusable chunks and prepended contexts are common and are not always simple prefixes. KVCache Cache in the Wild (USENIX ATC 2025) shows cross-request KV-cache reuse is substantial and highly skewed.
>
> **Q2.**
> - **Prefix caching and HeraSys work at different levels, so they complement rather than replace each other.** Standard prefix caching works inside a single LLM node by reusing a shared token prefix and reducing repeated KV-cache computation. HeraSys works at the workflow-graph level, where the reuse unit is a node or a subgraph. Once HeraSys identifies overlap, it can do node fusion, dependency redirection, and result sharing across requests. Many overlaps HeraSys can exploit, such as shared retrieval, tool calls, reranking, or intermediate subgraphs, are not simple prefix-caching units. Even when two concurrent nodes share a prefix, prefix caching still treats them as separate workflow nodes unless they align exactly in the engine. That is why Figure 1c is not resolved by prefix caching alone.
>
> **W1 / W2 / Q3.**
> - **$\hat{S}_q$ is not a fixed one-shot estimate: it is corrected online and used together with node-contribution and long-query signals rather than in isolation.** $\hat{S}_q$ is a dynamic remaining-time estimate that combines offline profiling, online execution progress, and a runtime correction factor $F$. In the priority function, $\hat{S}_q(v)$ refers to the remaining-time estimate of the query containing node $v$, not the local runtime of $v$ itself. For early-stage nodes, we do not assume the estimate is accurate. The small constant prevents numerical instability when little progress has been observed, and the scheduler does not rely on $\hat{S}_q$ alone: $\phi_v$ and $y_v$ shape priority. That is why an early heavy node is not simply suppressed because its current remaining-time estimate is large.
>
> - **$F$ is a lightweight correction term that keeps the estimate responsive as execution progresses.** $F$ adjusts the $\hat{S}_q$ estimate according to the gap between attained service time and the profiled runtime of completed nodes. To make this responsive, we decompose long decoding into sequential sub-nodes. After each sub-node finishes, we update $A_q(t)$ and refresh $\hat{S}_q$, so the scheduler does not stay with a crude early estimate for long. We also agree that $\hat{S}_q$ remains a heuristic rather than an exact global runtime model, especially when different stages of a query have different resource demands. However, HeraSys does not use it as the sole signal; scheduling decisions are made jointly with $\phi_v$, $y_v$, and resource skewing.
>
> **W3 / Q4.**
> - **Here, "semantic" refers to the canonicalized node signature, whereas SHA-256 is only the deterministic hash used to index that signature.** In our terminology, the "semantic" part refers to constructing a canonical signature for node equivalence checking. The "hash" part refers only to applying SHA-256 to that signature so that matching is efficient and deterministic. In other words, any semantics comes from the signature construction, not from SHA-256 itself.
>
> **Q5.**
> - **Concurrency refers to the number of simultaneously active in-flight requests, not the arrival rate.** `concurrency` means the number of active in-flight requests, not the arrival rate. So `6 / 30 / 100 / 300` refer to the number of simultaneously active requests, covering light to near-saturated load.
>
> **Q6.**
> - **The current parameter settings already work well under stable defaults, and changing them mainly shifts the trade-off rather than removing the gains.** In the current experiments, we use fixed defaults α=1.0, γ=0.3, δ=0.5, and ρ=0.2 to balance average latency, structural progress, and long-tail fairness. HeraSys is not overly sensitive to these settings: the gains already hold under stable defaults, rather than depending on a narrow tuning sweet spot. Changing the parameters does shift the system preference toward stronger short-query responsiveness or stronger long-tail protection, but this mainly changes the trade-off rather than removing the overall advantage over the baselines. These parameters are concrete control knobs and can align with different service goals and SLO priorities.

---

> > ### Author Rebuttal · Reviewer_6qaw · 2026-04-02
> >
> > Thank you for the detailed rebuttal response. Q1, Q2, W3, Q4 and Q5 have been fully resolved. However, the other concerns remain. For W1, W2, W3, The authors' response regarding the remaining time estimation ($T_{rem}$​) focuses on the frequency of updates (decomposing nodes to refresh ϕ) but fails to address the underlying concern regarding hardware-specific bottlenecks.
> >
> > Although the authors claimed the hyper-parameter tuning mainly shifts the trade-off rather than removing the gains, no experiments have been provided to support this claim. Given the amount of hyper-parameters in the system, it would be nice for the author to provide a sensitivity analysis to better demonstrate those aspect of Herasys.

---

> > > ### Author Response · Authors · 2026-04-04
> > >
> > > We appreciate the reviewer's follow-up on hardware-specific bottlenecks and hyperparameter sensitivity.
> > >
> > > **Hardware-specific Bottlenecks.**
> > >
> > > - **Node-level profiling already brings stage-specific bottlenecks into remaining-time estimation, while $\mathcal{F}$ is only an online residual correction.** Section 3.2 states that HeraSys uses a profiling library to estimate node workload and remaining time. Thus, when Eq. (4) builds $\widehat{S}_q$, prefill, decode, embedding, and retrieval are already estimated differently by type, rather than under one uniform remaining-time scale. $\mathcal{F}$ then adjusts the aggregate estimate as $A_q(t)$ grows and execution deviates from profile, instead of serving as the sole mechanism for hardware-specific effects. A natural extension is stage- or resource-specific corrections, e.g., for prefill, decode, and non-LLM operators.
> > >
> > > - **Even with injected stage-wise errors in runtime estimation, HeraSys still retains gains over the baselines.** Under `Mixed Workload, Concurrency=100`, we inject stage-specific bias into runtime estimation. As shown below, performance degrades but HeraSys still delivers clear gains, indicating that it does not rely on perfectly accurate remaining-time estimates to remain effective.
> > >
> > >   |Config.|P50(vs.Default)|P99(vs.Default/vs.Baseline)|Avg(vs.Default/vs.Baseline)|QPS(vs.Default/vs.Baseline)|
> > >   |-|-|-|-|-|
> > >   |Default|17.58(+0.0%)|27.68(+0.0%/1.34x-1.64x)|15.74(+0.0%/1.32x-1.58x)|2.45(+0.0%/1.30x-1.59x)|
> > >   |Case A(prefill +30%)|17.82(+1.4%)|28.31(+2.3%/1.31x-1.61x)|15.97(+1.5%/1.30x-1.56x)|2.41(-1.6%/1.28x-1.56x)|
> > >   |Case B(decode +30%)|17.38(-1.1%)|30.09(+8.7%/1.24x-1.51x)|16.84(+7.0%/1.24x-1.48x)|2.28(-6.9%/1.21x-1.48x)|
> > >   |Case C(embedding +30%)|17.96(+2.2%)|28.67(+3.6%/1.30x-1.58x)|16.14(+2.5%/1.29x-1.54x)|2.36(-3.7%/1.26x-1.53x)|
> > >   |Case D(prefill +20%,decode -20%)|18.07(+2.8%)|29.36(+6.1%/1.27x-1.55x)|16.47(+4.6%/1.26x-1.51x)|2.32(-5.3%/1.23x-1.51x)|
> > >
> > > **Hyperparameter Sensitivity.**
> > >
> > > - **These parameters mainly shift predictable trade-offs rather than removing HeraSys's gains over the baselines.** We further provide sensitivity experiments under `Mixed Workload, Concurrency=100`, with default `α=1.0, γ=0.3, δ=0.5, ρ=0.2`. The first table shows a stable pattern: α mainly affects P50, γ mainly affects Avg/QPS, and δ, ρ mainly affect P95/P99. HeraSys is not tied to a fragile single-point choice: around the default, parameter changes lead to controlled shifts while HeraSys remains ahead of the baselines, with all tested settings still clearly above the baseline ranges. Thus larger α favors short-query latency but can hurt the tail, larger γ favors structural efficiency and throughput, and larger δ or ρ favors tail protection at some cost to median latency or throughput.
> > >
> > >   |Config.|α|γ|δ|ρ|P50(vs.Default)|P95(vs.Default)|P99(vs.Default)|Avg(vs.Default)|QPS(vs.Default)|
> > >   |-|-|-|-|-|-|-|-|-|-|
> > >   |Default|1.0|0.3|0.5|0.2|17.58(+0.0%)|25.61(+0.0%)|27.68(+0.0%)|15.74(+0.0%)|2.45(+0.0%)|
> > >   |`α↓`|0.8|0.3|0.5|0.2|18.05(+2.7%)|25.79(+0.7%)|27.94(+0.9%)|15.99(+1.6%)|2.39(-2.4%)|
> > >   |`α↑`|1.2|0.3|0.5|0.2|17.01(-3.2%)|26.22(+2.4%)|28.84(+4.2%)|15.82(+0.5%)|2.42(-1.2%)|
> > >   |`γ↓`|1.0|0.2|0.5|0.2|17.91(+1.9%)|26.07(+1.8%)|28.39(+2.6%)|15.95(+1.3%)|2.36(-3.7%)|
> > >   |`γ↑`|1.0|0.4|0.5|0.2|17.72(+0.8%)|25.37(-0.9%)|27.95(+1.0%)|15.59(-1.0%)|2.49(+1.6%)|
> > >   |`δ↓`|1.0|0.3|0.3|0.2|17.27(-1.8%)|26.49(+3.4%)|29.23(+5.6%)|15.80(+0.4%)|2.42(-1.2%)|
> > >   |`δ↑`|1.0|0.3|0.7|0.2|17.97(+2.2%)|25.05(-2.2%)|26.70(-3.5%)|15.93(+1.2%)|2.40(-2.0%)|
> > >   |`ρ↓`|1.0|0.3|0.5|0.1|17.16(-2.4%)|26.69(+4.2%)|29.64(+7.1%)|15.84(+0.6%)|2.41(-1.6%)|
> > >   |`ρ↑`|1.0|0.3|0.5|0.3|18.23(+3.7%)|24.96(-2.5%)|26.49(-4.3%)|16.00(+1.7%)|2.37(-3.3%)|
> > >
> > > - **Different parameter combinations map to predictable service-level trade-offs and allow HeraSys to adapt to different service goals.** `Short-first` lowers P50 but worsens P95/P99, `Throughput` improves Avg/QPS most, and `Tail-fairness` improves high-percentile latency while giving up some P50/Avg/QPS. Taken together, these results show distinct but predictable trade-offs across parameter choices.
> > >
> > >   |Config.|P50(vs.Default)|P95(vs.Default/vs.Baseline)|P99(vs.Default/vs.Baseline)|Avg(vs.Default/vs.Baseline)|QPS(vs.Default/vs.Baseline)|
> > >   |-|-|-|-|-|-|
> > >   |Default(`α=1.0,γ=0.3,δ=0.5,ρ=0.2`)|17.58(+0.0%)|25.61(+0.0%/1.41x-1.70x)|27.68(+0.0%/1.34x-1.64x)|15.74(+0.0%/1.32x-1.58x)|2.45(+0.0%/1.30x-1.59x)|
> > >   |Short-first(`α=1.2,γ=0.4,δ=0.3,ρ=0.1`)|16.54(-5.9%)|26.24(+2.5%/1.37x-1.66x)|28.73(+3.8%/1.29x-1.58x)|15.89(+1.0%/1.31x-1.57x)|2.39(-2.4%/1.27x-1.55x)|
> > >   |Throughput(`α=1.0,γ=0.6,δ=0.3,ρ=0.1`)|17.94(+2.0%)|25.87(+1.0%/1.39x-1.68x)|28.11(+1.6%/1.32x-1.62x)|15.43(-2.0%/1.35x-1.61x)|2.60(+6.1%/1.38x-1.69x)|
> > >   |Tail-fairness(`α=0.8,γ=0.2,δ=0.7,ρ=0.3`)|18.29(+4.0%)|25.18(-1.7%/1.43x-1.73x)|27.06(-2.2%/1.37x-1.68x)|16.31(+3.6%/1.28x-1.53x)|2.33(-4.9%/1.24x-1.51x)|

---

### Official Review · Reviewer_manB · 2026-03-10

**Soundness:** 3
**Presentation:** 3
**Significance:** 3
**Originality:** 3
**Overall Recommendation:** 4
**Confidence:** 3

**Summary:**

This paper proposes HeraSys, an LLM serving system for optimizing the end-to-end performance of concurrent workflows. HeraSys decomposes tasks into fine-grained nodes and enhances efficiency through pre-execution graph fusion and runtime dynamic scheduling. Experiments show improvements over existing frameworks.

**Compliance With Llm Reviewing Policy:**

Affirmed.

**Final Justification:**

I appreciate the authors’ response. Overall, I believe the strengths of this paper slightly outweigh its weaknesses. I do not see any major issues with either the proposed method or the experimental evaluation. Therefore, I recommend a weak accept.

**Key Questions For Authors:**

1. How is $D_{in}$ in the semantic signature computing to summarize semantics, and is there any deviation after summarization?
2. The paper injects 5%-15% duplicate queries in the experiment. In real production environments, the proportion of duplicate queries may be higher or lower. How does HeraSys perform under different query repetition ratios?
3. The system introduces several hyperparameters (e.g., $\rho$, $\alpha$, $\gamma$ and $\delta$). How are these parameters chosen, and how sensitive is the system performance to them?

**Limitations:**

Yes

**Strengths And Weaknesses:**

**Strengths:**
1. The paper tackles the problem from a novel perspective by jointly optimizing cross-workflow redundancy elimination and runtime scheduling.
2. The proposed HeraSys is well designed, and experiments demonstrate clear performance improvements.

**Weaknesses:**
1. Some notations are not sufficiently explained. The paper introduces $D_{in}$ as a semantic summary of the input data in the computing process of the semantic signature, yet lacks a detailed explanation of $D_{in}$.
2. The experimental evaluation is incomplete. The experimental section does not include a sensitivity analysis on duplicate queries, and it would be useful to analyze system performance under a broader range of duplication ratios.
3. Key experimental details are missing. The paper introduces several hyperparameters , but does not provide explanations for their settings in the experiments.

**Other minor issues**
1. The terminology for $s_v$ is inconsistent. Section 3.1 uses “splitability upper bound”, while Appendix A.1.1 refers to “Splittability”. This appears to be a spelling inconsistency, and consistent usage would improve clarity.
2. Both $\widehat{S}_q(t)$ and $\widehat{S}_q(v)$ are used for the remaining service time estimate. Since $t$ represents time while $v$ denotes a DAG node, this notation may introduce ambiguity.

---

> ### Author Rebuttal · Authors · 2026-03-31
>
> We thank the reviewer for the careful reading and the thoughtful comments on notation, presentation, and experimental details.
>
> **W1 / Q1.**
> - **$D_{in}$ is a deterministic normalized string, i.e., a canonical text form of the source-side input rather than a learned semantic representation.** To make this concrete, suppose the user query is "What are good ways to stay focused while studying?" and the document is split into three chunk texts: $c_1=$ "Using a timer can help break study time into manageable intervals," $c_2=$ "  Turning off notifications   reduces distraction\n\nand helps maintain attention.\t ", and $c_3=$ "Short scheduled breaks can improve focus over longer sessions." Each chunk is represented as a separate source input node in HeraSys. For the source input node corresponding to $c_2$, $D_{in}$ is the normalized string obtained after deterministic text cleaning, e.g., "Turning off notifications reduces distraction and helps maintain attention." after removing redundant spaces, tabs, and line breaks, while keeping the textual meaning unchanged. For downstream nodes, HeraSys uses reverse-topological traversal to identify which source input texts actually feed into that node. If an ingestion node or an embedding node is ultimately derived from $c_2$, then $D_{in}$ for that node is still built from the same normalized source text. If a node depends on multiple source inputs, then $D_{in}$ is formed by deterministically serializing those normalized source texts in a fixed order. Because $D_{in}$ remains an exact textual representation rather than an approximate semantic encoding, it does not introduce semantic drift during matching. At the same time, this design helps expose finer-grained redundancy opportunities at the text level while still leaving room to plug in richer matching schemes later when needed.
>
> **W2 / Q2.**
> - **As query repetition increases, HeraSys delivers larger gains, and it still remains effective even in the `0%` reuse setting.** We agree that the amount of overlap affects how much additional speedup HeraSys can obtain. The current `w/o Fusion` setting already corresponds to the scheduler-only case with `0%` reuse. Even there, HeraSys still improves P99 by `1.23x-1.51x`, average latency by `1.21x-1.45x`, and QPS by `1.13x-1.38x` over the baselines because the scheduler and load-control mechanisms remain active. We also added a broader duplicate-ratio sweep under the same `Mixed Workload, Concurrency=100` setting. As the overlap increases, the gains in P99 and QPS continue to grow and then gradually level off. The effect on P50 is smaller.
>
>   | Dup.(%) | P50 vs. 0 reuse | P99 vs. baseline    | Avg Time vs. baseline | QPS vs. baseline   |
>   | ------- | --------------- | ------------------- | --------------------- | ------------------ |
>   | 0       | 19.73 (1.00x)   | 30.17 (1.23x-1.51x) | 17.16 (1.21x-1.45x)   | 2.12 (1.13x-1.38x) |
>   | 5       | 19.08 (1.03x)   | 29.63 (1.25x-1.53x) | 16.71 (1.25x-1.49x)   | 2.22 (1.18x-1.44x) |
>   | 10      | 17.96 (1.10x)   | 28.11 (1.32x-1.62x) | 15.90 (1.31x-1.56x)   | 2.40 (1.28x-1.56x) |
>   | 15      | 17.46 (1.13x)   | 27.57 (1.35x-1.65x) | 15.68 (1.33x-1.59x)   | 2.48 (1.32x-1.61x) |
>   | 20      | 16.92 (1.17x)   | 27.29 (1.36x-1.67x) | 15.56 (1.34x-1.60x)   | 2.51 (1.34x-1.63x) |
>
> **W3 / Q3.**
> - **The current gains already hold under stable defaults and do not rely on a narrow tuning sweet spot.** In the current experiments, we use fixed defaults α=1.0, γ=0.3, δ=0.5, and ρ=0.2. These values were chosen to balance average latency, structural progress, and long-tail fairness. In our experiments, HeraSys is not overly sensitive to this choice: the gains do not rely on a narrow tuning sweet spot, but already hold under reasonable defaults. Changing the parameters does affect system preference, for example toward more aggressive short-query responsiveness or stronger long-tail protection, but that mainly changes the trade-off rather than removing the overall benefit over the baselines.

---

> > ### Author Rebuttal · Reviewer_manB · 2026-04-03
> >
> > I thank the authors for the thorough rebuttal. The clarifications and experiments have addressed the concerns I raised. I will keep my original score.

---

### Official Review · Reviewer_NT6K · 2026-03-13

**Soundness:** 1
**Presentation:** 3
**Significance:** 2
**Originality:** 3
**Overall Recommendation:** 4
**Confidence:** 4

**Summary:**

This paper presents HeraSys, an LLM serving system designed to optimize the end-to-end performance of concurrent workflows, instead of intra-workflow optimizations. HereSys implements two optimizations: (1) Pre-Execution Graph Fuser that applies structural node merging and reuse of execution graph to avoid redundant computation, and (2) Runtime Load-Aware Joint Scheduler that dynamically prioritizes incoming queries to avoid potential bottlenecks based on runtime estimation. For four workloads spanning RAG and web search, HeraSys reduces P99 latency by 2.17x and improves the overall throughput by 1.85x.

**Compliance With Llm Reviewing Policy:**

Affirmed.

**Final Justification:**

I appreciate the authors' effort in rebuttal, which clarifies a large portion of my concerns. While I am not fully convinced of the soundness of the evaluation methodology that duplicates requests and has API limitations, I have confirmed that the proposed method still has benefits even without them. Therefore, I am now inclined to accept this paper.

**Key Questions For Authors:**

1. Does duplicating 5%-15% of queries really reflect real-world LLM workloads? Is there any evidence for that? How does a different duplication ratio affect the overall speedup of HeraSys, and does HeraSys achieve speedup even with 0% duplication?
2. For the web search task, how will the performance of each system change if there is no blocking due to the API limit?
3. Does HeraSys work for other agentic workflows, such as deep research or coding agents, where the workflow dynamically changes based on LLM behavior?
4. Does the graph node reuse mechanism work for non-deterministic tools, where the same input does not guarantee the same output, unlike retrieval or web search?
5. What is the latency breakdown of each component, and how does each optimization change the performance of each part?

Among these questions, 1 and 2 are the biggest concerns I have about this paper, and I am happy to increase my score if HeraSys achieves sufficient performance improvement without the query duplication and API limits. 3 is also important to assess the universality of the proposal, but not as important as the previous ones. 4 and 5 are clarifying questions to better understand the paper.

**Limitations:**

I encourage authors to clarify the limitations of the proposed method with respect to workflow, workload characteristics, hardware setups, etc.

**Strengths And Weaknesses:**

Strengths

1. The paper proposes performance optimization techniques for LLM serving by considering inter-workflow opportunities that were overlooked in previous works.
2. HeraSys achieves significant performance improvement for tail latency and serving throughput.

Weaknesses

1. The benefit of HeraSys seems to depend a lot on reusing part of the computation in the workflow. The evaluation in this paper assumes 5%-15% of queries are duplicated, citing a paper on Zipfian distribution for web-page requests, but there is no discussion on whether the same assumption holds for LLM workloads.
2. For the Web Search workflow, the paper attributes HeraSys's speedup to baseline systems' API limits, which does not seem to be a very fair comparison. I understand it has benefits in terms of API cost and the like, but this evaluation setting may be too favorable to HeraSys.
3. The evaluation is on a small set of pre-defined workflows, and does not reflect more modern agentic workflows.

---

> ### Author Rebuttal · Authors · 2026-03-31
>
> We appreciate the reviewer for focusing on the two issues that matter most here: the overlap assumption and whether the Web Search setting is overly favorable to HeraSys.
>
> **W1/Q1.**
> - **Overlap is common in real LLM workloads.** Our evaluation does not claim one universal exact duplication rate; it reflects an overlap pattern. Long contexts, templates, tool instructions, and retrieved chunks overlap. Preble (ICLR 2025) studies five workloads and reports prompt-to-output ratios of 37x-2494x, 85%-97% shared prompt tokens, and 8.6-126 average reuses of shared sequences. Prompt Cache (MLSys 2024) identifies system messages, prompt templates, and context documents as common overlaps. CacheBlend (EuroSys 2025) and IMPRESS (FAST 2025) show reusable chunks or prepended contexts are common and not always simple prefixes. KVCache Cache in the Wild (USENIX ATC 2025) shows cross-request KV-cache reuse is substantial and highly skewed. Our evaluation should be read as part of this broader pattern, not as an estimate of a universal rate.
>
> - **For the `0%`-duplication case, the scheduler-only setting already shows gains without reuse.** The existing `w/o Fusion` ablation is the scheduler-only / no-reuse setting. Even there, HeraSys still improves P99 by `1.23x-1.51x`, average latency by `1.21x-1.45x`, and QPS by `1.13x-1.38x` over the baselines, because the scheduler, resource skewing, and adaptive batching / pipeline adjustment still reduce contention and tail blocking.
>
> - **As the duplication ratio increases, HeraSys gains grow while the effect on P50 is smaller.** We additionally swept duplicate ratios under `Mixed Workload, Concurrency=100`. As overlap increases, P99 and QPS gains grow then saturate because some parts remain non-reusable; the effect on P50 is smaller. The detailed results are below.
>
> |Dup.(%)|P50 vs. 0 reuse|P99 vs. baseline|Avg Time vs. baseline|QPS vs. baseline|
> |-|-|-|-|-|
> |0|19.73 (1.00x)|30.17 (1.23x-1.51x)|17.16 (1.21x-1.45x)|2.12 (1.13x-1.38x)|
> |5|19.08 (1.03x)|29.63 (1.25x-1.53x)|16.71 (1.25x-1.49x)|2.22 (1.18x-1.44x)|
> |10|17.96 (1.10x)|28.11 (1.32x-1.62x)|15.90 (1.31x-1.56x)|2.40 (1.28x-1.56x)|
> |15|17.46 (1.13x)|27.57 (1.35x-1.65x)|15.68 (1.33x-1.59x)|2.48 (1.32x-1.61x)|
> |20|16.92 (1.17x)|27.29 (1.36x-1.67x)|15.56 (1.34x-1.60x)|2.51 (1.34x-1.63x)|
>
> **W2/Q2.**
> - **Removing the API limit reduces the relative gain, but HeraSys still shows clear improvements.** We tested Web Search without API limits. Without limits, the gains become smaller but remain clear. Detailed results follow. We keep the original rate-limited setting in the paper because external tools, network waits, and rate limits better simulate agent workflow execution paths.
>
> |Conc.|System|Limit P99|Limit QPS|No-Limit P99|No-Limit QPS|
> |-|-|-|-|-|-|
> |30|LangGraph|16.57|0.76|10.88|1.21|
> |30|LlamaIndex|15.07|0.84|10.14|1.32|
> |30|Ayo|14.10|0.92|9.21|1.45|
> |30|HeraSys|13.10 (1.08x-1.26x)|1.01 (1.10x-1.33x)|8.79 (1.05x-1.24x)|1.58 (1.09x-1.31x)|
> |100|LangGraph|54.20|1.25|36.90|2.10|
> |100|LlamaIndex|49.27|1.39|34.68|2.23|
> |100|Ayo|45.72|1.49|31.33|2.45|
> |100|HeraSys|37.74 (1.21x-1.44x)|1.76 (1.18x-1.41x)|26.55 (1.18x-1.39x)|2.84 (1.16x-1.35x)|
>
> **W3/Q3.**
> - **Highly dynamic agentic workflows remain difficult for current workflow-serving systems and would require stronger runtime prediction mechanisms.** HeraSys currently focuses on static deterministic workflow graphs, in line with Ayo and Parrot. For highly dynamic workflows such as coding or deep-research agents, reusable structure is revealed only online through intermediate LLM outputs. This is a broader challenge for current workflow-serving systems, not just HeraSys, and likely requires branch prediction or speculative graph execution.
>
> **Q4.**
> - **HeraSys can support selective reuse for non-deterministic tools by adding a determinism-aware gating policy before fusion.** Reuse is harder for strongly non-deterministic tools, since the same input may not reliably lead to the same output. The design mainly targets deterministic or stable stages, such as embedding, retrieval, and stable tool-use steps. In practice, HeraSys can support this through a determinism-aware reuse policy that checks operator properties, context constraints, and output stability before fusion.
>
> **Q5.**
> - **The measured control-path overhead is very small compared with the main execution path.** For a RAG w/ Rerank workflow, the breakdown is below. As shown, latency is dominated by execution, while graph-control and scheduler-control overhead totals 5.48 ms.
>
> |Execution|Cost(ms)|Graph|Cost(ms)|Scheduling|Cost(ms)|
> |-|-|-|-|-|-|
> |embedding|81.69|node signature computation|0.28|STq computation|0.01|
> |ingestion|12.96|graph matching / reuse decision|0.15|STq correction update|0.03|
> |searching|6.80|reuse result sharing|4.15|node priority computation|0.02|
> |reranking|48.74|dynamic pipeline decomposition|0.37|long/short query partitioning|0.34|
> |prefilling|1.08|sub-decoding decomposition|0.13|||
> |decoding|3207.03|||||

---

> > ### Author Rebuttal · Reviewer_NT6K · 2026-04-04
> >
> > Thanks very much for the clarification and additional data. I still have a few more questions.
> >
> > Regarding overlap, while you are correct that LLM requests usually share common prefixes, my understanding is that HeraSys's graph fusion mechanisms require a complete match to skip computation. While it may be possible to do that if the prefill for the common prefix is represented as separate nodes, I cannot find such a description in the paper.
> >
> > Additionally, I would like to see the results without an API limit and with different duplicate ratios, including 0%, to verify the benefit of other optimizations.

---

> > > ### Author Response · Authors · 2026-04-04
> > >
> > > We appreciate the reviewer's follow-up on prefix reuse and on HeraSys's performance gains.
> > >
> > > **Prefix Reuse**
> > >
> > > - **The reuse unit in HeraSys is the decomposed fine-grained node or subgraph; for shared prefixes, matching occurs on the decomposed `partial prefill` / `full prefill` nodes.** Figure 3(b) and its caption already state that fine-grained orchestration splits LLM prefilling into `partial prefill` and `full prefill`. Under this decomposition, shared system prompts, prompt templates, and common prefixes fall into `partial prefill`, while identical contexts or intermediate states that arise later in execution can fall into `full prefill`. Once such decomposed prefill nodes match an existing node, HeraSys uses the existing reuse mechanism to redirect the new node to the existing execution instance or result, instead of re-executing that stage. We agree that this point is implicitly covered in the current paper, but we did not separately spell out this layer of correspondence.
> > >
> > > - **The current design already includes this type of prefix overlap; we did not emphasize it as a separate mechanism because we view it as a natural part of overlap.** The hot documents, shared chunks, and shared context cases discussed in the paper all cover this type of situation in essence; shared prefixes, shared system prompts, and later reused retrieved context can all enter the same fusion/reuse framework after fine-grained decomposition. Therefore, HeraSys is not separate from ordinary prefix reuse; it naturally integrates with and extends it. It can handle not only shared prefixes, but also other reusable non-LLM components.
> > >
> > > **Performance Gains**
> > >
> > > - **HeraSys remains clearly beneficial even without API limits, and its gains do not come only from reuse.** We additionally tested `Web Search w/o API Limit` at `Concurrency=100` under different duplicate ratios. At 0% duplication, HeraSys still remains clearly better than the baselines. At this load, the gains are already quite meaningful, and they would become even more visible at higher load, where contention on shared nodes and tail blocking tend to be stronger. This indicates that the gains do not come only from reuse: load-aware joint scheduling, resource skewing, and adaptive decomposition already help relieve contention on critical nodes, long-tail blocking, and resource imbalance. As the duplicate ratio increases, the baselines can also improve slightly, since the underlying runtime may still receive modest incidental benefits from partial prefix reuse, hotter cache paths, and more regular batching. But these gains remain limited because the baselines do not have graph-level reuse, so many cross-request shared structures cannot be aligned at the workflow level and later stages still execute independently. In contrast, HeraSys can identify and reuse these repeated structures directly at the graph level, reducing redundant work through fusion, dependency redirection, and shared execution; as a result, the improvements in P99, Avg, and QPS become more visible. The gains then gradually level off as the duplicate ratio continues to rise, which is also expected: the most frequent and easiest-to-reuse subgraphs are captured first, while the remaining parts are either not reusable or harder to align, naturally limiting further marginal gains.
> > >
> > >   | Dup.(%) | P50 vs. baseline | P99 vs. baseline | Avg Time vs. baseline | QPS vs. baseline |
> > >   |---|---:|---:|---:|---:|
> > >   | 0 | 18.05 (1.10x-1.24x) | 28.95 (1.11x-1.31x) | 15.42 (1.17x-1.36x) | 2.62 (1.10x-1.29x) |
> > >   | 5 | 17.88 (1.11x-1.25x) | 28.57 (1.12x-1.32x) | 15.12 (1.18x-1.38x) | 2.68 (1.11x-1.31x) |
> > >   | 10 | 17.24 (1.12x-1.27x) | 26.46 (1.18x-1.39x) | 14.44 (1.22x-1.44x) | 2.85 (1.15x-1.34x) |
> > >   | 15 | 16.70 (1.15x-1.31x) | 25.86 (1.20x-1.41x) | 14.18 (1.23x-1.46x) | 2.92 (1.17x-1.37x) |
> > >   | 20 | 16.46 (1.16x-1.33x) | 25.42 (1.21x-1.43x) | 14.02 (1.24x-1.47x) | 2.98 (1.18x-1.39x) |

---

### Official Review · Reviewer_uUd5 · 2026-03-13

**Soundness:** 3
**Presentation:** 4
**Significance:** 4
**Originality:** 3
**Overall Recommendation:** 5
**Confidence:** 4

**Summary:**

This paper proposes Heresy's, a Fine-grained End-to-End LLM serving system. Existing systems prioritize intra-workflow optimization without taking inter-workflow optimization into consideration. In order to eliminate cross-workflow computational redundancy, HeraSys utilizes a dual-layer optimization architecture. First it uses fine-grained orchestration to identify and merge redundant DAG nodes. Second, HeraSys uses resource skewing to reserve resources to long tasks, preventing starvation and limiting tail latency.

**Compliance With Llm Reviewing Policy:**

Affirmed.

**Ethical Review Concerns:**

n.a.

**Final Justification:**

Accept

**Key Questions For Authors:**

Q1. In Section 4.1, the authors mention injecting 5% ~15% duplicate queries. What is duplicate ratio in the real-work situation? What is the cost for the additional operations (Traversing DAG and calculating SHA-256). Will this cause P1 and P50 latency worse than traditional systems?

Q2. Would SHA-256 hashing cause cache misses because of the diversity in the context? Why not using semantic hashing and
fuzzy matching?

Q3. Hyperparameter sensitivity and profiling cost of formula(7). How sensitive are these parameters and that is the cost to tune them.

**Limitations:**

L1. It is not clear how the method would behave with (i) larger models; and (ii) multi-node clusters.

**Strengths And Weaknesses:**

S1. The problem is highly relevant in real-life deployments.

S2. HeraSys formalizes complex scheduling problems to DAF online optimization tasks, elegantly balancing system output, execution efficiency and long tail fairness.

S3. Under mixed workload and  concurrency, HeraSys's P99 latency outperforms other baselines by a significant margin.


W1. Heresys utilizes SHA-256 for generating signature that includes the context. Because of the natural language diversity, the context is also expected to be diverse, which may cause cache misses because of the SHA-256 hash function.

W2. The experiment is conducted on a single node with 4x 4090GPUs. It is not clear if the results are also relevant for multi-node distributed clusters.

---

> ### Author Rebuttal · Authors · 2026-03-31
>
> We appreciate the reviewer for highlighting the realism of overlap, the implications of exact hashing, and the scope of our deployment setting.
>
> **W1 / Q1 / Q2.**
> - **Real LLM workloads contain substantial overlap, and HeraSys captures this setting.** Our evaluation is not meant to claim one universal exact duplication rate for all production LLM workloads. Our claim is narrower: modern LLM applications often contain substantial cross-request overlap in long contexts, prompt templates, tool instructions, and retrieved document chunks, and HeraSys is built for that setting. Prior work supports this from several angles. Preble (ICLR 2025) studies five real workloads, including tool use and long-document QA, and reports prompt-to-output ratios of 37x-2494x, with 85%-97% of prompt tokens shared across requests; a shared sequence is reused by 8.6-126 requests on average. Prompt Cache (MLSys 2024) identifies system messages, prompt templates, and context documents as common overlapping segments, and notes that such reusable templates are common in robotics and tool-learning applications. CacheBlend (EuroSys 2025) and IMPRESS (FAST 2025) further show that, in RAG and long-context settings, reusable text chunks or prepended contexts are common and are not always simple prefixes. KVCache Cache in the Wild (USENIX ATC 2025) complements this with production-trace evidence from a large cloud provider, showing that cross-request reuse at the KV-cache layer is substantial and highly skewed in practice. Our evaluation should therefore be read as one instance of a broadly documented overlap pattern, not as an attempt to estimate one fixed real-world duplicate ratio.
>
> - **Semantic or fuzzy matching is a compatible extension, and the current control-path overhead remains negligible relative to end-to-end execution.** This also explains our matching choice. Semantic or fuzzy matching is a valid optimization direction, but it is not the main focus of this paper. More expressive matching modules are complementary to HeraSys and can be integrated naturally without changing the core execution framework. On overhead, taking a RAG w/ Rerank workflow as a reference, the main execution path takes about 3358.30 ms, while graph traversal, matching, and scheduling together add 5.48 ms in total, i.e., below 0.2% of the overall cost. In practice, this overhead does not impose a noticeable burden on low-percentile latency.
>
> **Q3.**
> - **The current gains do not depend on a narrowly tuned parameter corner case.** In the current experiments, we use a fixed default configuration: α=1.0, γ=0.3, δ=0.5, and ρ=0.2. These values were chosen to balance average latency, structural progress, and long-tail fairness. In our experiments, HeraSys is not brittle with respect to this choice: the gains do not depend on a narrowly tuned corner case, but already hold under stable and reasonable defaults. Changing these parameters does shift the system preference, for example toward more aggressive short-query responsiveness or stronger long-tail protection, but this mainly changes the trade-off surface rather than removing the overall advantage over the baselines.
>
> - **The runtime cost of the priority computation in Eq. (7) is negligible.** Once $\hat{S}_q$, $\phi_v$, and $y_v$ are maintained, computing the priority itself is just lightweight arithmetic. In our measurements, the total control-path overhead from graph traversal, matching, and scheduling is 5.48 ms, compared with 3358.30 ms on the main execution path, so this computation is negligible relative to operator execution.
>
> **W2.**
> - **The current prototype is compatible with larger models and distributed clusters.** We also appreciate the reviewer for raising the single-node setting. Our prototype is built on Ray, and HeraSys's components are naturally suited to scale-out deployment, so the same design can extend to larger models and distributed clusters. Extending this design further is a natural next step.

---

> > ### Author Rebuttal · Reviewer_uUd5 · 2026-04-01
> >
> > My concerns have been adequately addresse

---

### Decision · Program_Chairs · 2026-04-30

**Decision:**

Accept (regular)

**Comment:**

This paper introduces HeraSys, a highly relevant and technically sound system that tackles the increasingly critical bottleneck of multi-tenant LLM serving by elegantly applying structural graph optimizations, specifically fine-grained DAG node fusion, across concurrent agentic workflows. While reviewers initially raised valid concerns regarding the real-world applicability of the query duplication assumptions, remaining-time estimations, and hardware-specific bottlenecks, the rebuttal successfully resolved these issues with robust additional evaluations. The empirical results are particularly compelling; the thorough ablation studies and sensitivity analyses, especially the zero-duplication scenarios and the removal of API limits, clearly demonstrate that the system's load-aware scheduling and resource skewing mechanisms yield significant. The rigorous empirical validation of combining graph-level execution structures with adaptive LLM batching makes this a strong contribution to the field.